# The RNA export and RNA decay complexes THO and TRAMP prevent transcription-replication conflicts, DNA breaks, and CAG repeat contractions

Rebecca E. Brown[1]☯, Xiaofeng A. Su[2,3]☯, Stacey Fair[2], Katherine Wu[2], Lauren Verra[2], Robyn Jong[2], Kristin Andrykovich[2], Catherine H. Freudenreich[1,2]*

1 Program in Genetics, Tufts University School of Graduate Biomedical Sciences, Boston, Massachusetts, United States of America, 2 Department of Biology, Tufts University, Medford, Massachusetts, United States of America, 3 David H. Koch Institute for Integrative Cancer Research, Department of Biology, Massachusetts Institute of Technology, Cambridge, Massachusetts, United States of America

☯ These authors contributed equally to this work.
* catherine.freudenreich@tufts.edu

**Data Availability Statement:** All relevant data are within the paper and its Supporting Information files. In addition, a detailed protocol for the

## Abstract

Expansion of structure-forming CAG/CTG repetitive sequences is the cause of several neurodegenerative disorders and deletion of repeats is a potential therapeutic strategy. Transcription-associated mechanisms are known to cause CAG repeat instability. In this study, we discovered that Thp2, an RNA export factor and member of the THO (suppressors of transcriptional defects of *hpr1Δ* by overexpression) complex, and Trf4, a key component of the TRAMP (Trf4/5-Air1/2-Mtr4 polyadenylation) complex involved in nuclear RNA polyadenylation and degradation, are necessary to prevent CAG fragility and repeat contractions in a *Saccharomyces cerevisiae* model system. Depletion of both Thp2 and Trf4 proteins causes a highly synergistic increase in CAG repeat fragility, indicating a complementary role of the THO and TRAMP complexes in preventing genome instability. Loss of either Thp2 or Trf4 causes an increase in RNA polymerase stalling at the CAG repeats and other genomic loci, as well as genome-wide transcription-replication conflicts (TRCs), implicating TRCs as a cause of CAG fragility and instability in their absence. Analysis of the effect of RNase H1 overexpression on CAG fragility, RNAPII stalling, and TRCs suggests that RNAPII stalling with associated R-loops are the main cause of CAG fragility in the *thp2Δ* mutants. In contrast, CAG fragility and TRCs in the *trf4Δ* mutant can be compensated for by RPA overexpression, suggesting that excess unprocessed RNA in TRAMP4 mutants leads to reduced RPA availability and high levels of TRCs. Our results show the importance of RNA surveillance pathways in preventing RNAPII stalling, TRCs, and DNA breaks, and show that RNA export and RNA decay factors work collaboratively to maintain genome stability.

detection of transcription-replication conflicts in yeast by proximity ligation assay (PLA) is available at DOI: dx.doi.org/10.17504/protocols.io.kxygx93rzg8j/v1.

**Funding:** National Science Foundation MCB, grant 1330743, awarded to C.H.F. and National Science Foundation MCB, grant 1817499, awarded to C.H. F. https://www.nsf.gov/div/index.jsp?div=MCB The funders had no role in study design, data collection and analysis, decision to publish, or preparation of the manuscript.

**Competing interests:** The authors have declared that no competing interests exist.

**Abbreviations:** ChIP, chromatin immunoprecipitation; CUT, cryptic unstable transcript; DRIP, DNA:RNA immunoprecipitation; DSB, double-strand break; HD, Huntington's disease; IP, immunoprecipitation; mRNP, messenger ribonucleoprotein; MSS-MLE, Ma-Sandri-Sarkar Maximum Likelihood Estimator; PLA, proximity ligation assay; RNAPII, RNA polymerase II; rRFB, rDNA replication fork barrier; ssDNA, single-stranded DNA; TAR, transcription-associated recombination; TCR, transcription-coupled repair; TNR, trinucleotide repeat; TRC, transcription-replication conflict; YAC, yeast artificial chromosome; YC, yeast complete; 5-FOA, 5-fluoroorotic acid.

## Introduction

Expansion-prone trinucleotide repeats (TNRs) are prone to DNA secondary structure formation, which can cause roadblocks to transcription or replication and interfere with DNA repair to cause repeat instability (changes in repeat length) and fragility (chromosome breaks) [1,2]. Transcription through TNRs is an important *cis*-acting factor increasing repeat instability, and the known expansion-prone TNRs are transcribed within their associated genes [1,3]. Tissue-specific RNA polymerase II (RNAPII) occupancy in the striatum and cerebellum during transcription elongation is highly associated with CAG repeat instability levels in Huntington's disease (HD) mouse models [4,5]. Transcription-coupled repair (TCR) has been reported to cause TNR instability in some systems [6]. However, TCR does not appear to be the major pathway causing expansions at the HD or Fragile X loci [1]. Therefore, other mechanisms of transcription-induced DNA damage and repair are likely also relevant.

Transcription through CAG and CGG repeats promotes R-loop formation and increases R-loop-dependent repeat instability [7–10]. R-loops promote single-stranded DNA (ssDNA) on the non-template strand, which is then available for secondary structure formation, and may provide a target for MutLγ nuclease cleavage and base excision repair [7,11]. Therefore, formation of stable R-loops is one way in which transcription through G-rich repeats can cause chromosome fragility and instability. Another way that transcription can induce repeat instability is through changes in DNA supercoiling or chromatin structure that allow formation of secondary DNA structures. For example, remodeling by Isw1 during transcription is important in preventing CAG repeat expansions by helping to reestablish histone spacing after passage of RNAPII [12]. In addition, CAG or CTG slip-out structures can cause transcriptional arrest [13–15]. Despite the known importance of transcription in causing TNR instability, the impact of transcription-coupled RNA processing pathways on CAG repeat instability and fragility remains mostly unknown.

The yeast THO complex is a conserved eukaryotic transcription elongation factor that interacts with the nuclear pore-associated TREX complex to facilitate mRNA export [16]. THO is composed of 4 proteins, Tho2, Hpr1, Mft1, and Thp2, which form a highly stable complex, and Tex1, which is less tightly associated [17–19]. The THO complex travels co-transcriptionally with RNAPII to ensure stable messenger ribonucleoprotein (mRNP) formation and RNA extrusion during transcription [20,21]. The THO-defective *hpr1Δ* mutant was shown to exhibit increased R-loop formation and transcription-associated recombination (TAR) [22]. Overexpression of RNase H1, the ribonuclease that directly degrades the RNA moiety in an RNA:DNA hybrid [23], abolishes the TAR phenotype in the *hpr1Δ* mutant [22]. THO mutants have a genome-wide accumulation of the Rrm3 protein, known to be required for replication through obstacles, in RNAPII transcribed genes [24–26]. The THO complex also counteracts telomere shortening and telomeric R-loop formation [27] and is needed for proper transcriptional elongation through the gene *FLO11*, which has internal tandem repeats [28]. Therefore, the THO complex could play an important role in maintaining stability and reducing DNA breaks within CAG repeats by facilitating mRNA export and preventing R-loops to ensure normal transcription elongation.

The TRAMP (Trf4/5-Air1/2-Mtr4 polyadenylation) complex is a functionally conserved nuclear RNA processing, degradation, and surveillance factor that facilitates degradation of RNA substrates by the addition of short unstructured poly(A) tails that target them for nuclear exosome-mediated degradation [29,30]. TRAMP complexes are composed of the RNA helicase Mtr4, one of the non-canonical poly(A) polymerases Trf4 or Trf5 (forming TRAMP4 or TRAMP5 complexes, respectively), and an RNA-binding protein, either Air1 or Air2 [30]. Although TRAMP4 and TRAMP5 have similarity in their structures and RNA polyadenylation

function, they also have specificity in RNA substrate species [30,31]. The TRAMP complexes are important for degradation of rRNAs, tRNAs, small nuclear/small nucleolar (sn/sno) RNAs, and cryptic unstable transcripts (CUTs) by the exosome complex [31,32]. In vitro studies have shown that the TRAMP adenylation and helicase activities act in a cooperative manner to unwind structured RNAs [30,33]. The TRAMP complex has also been found to be cotranscriptionally recruited to promote rapid degradation of unwanted RNA transcripts, including spliced out introns, cryptic transcripts from rDNA regions, and aberrant mRNPs [32,34,35]. Deletion of yeast Trf4 causes nascent mRNA-mediated TAR, terminal deletions, and chromosome loss, which is suppressed by overexpression of RNase H1 [36,37]. TAR is also evident in mutants lacking Rrp6, a major exoribonuclease of the nuclear exosome [33,38], and *rrp6Δ* mutants were shown to be associated with R-loop formation [39,40]. Therefore, emerging evidence has revealed new links between the TRAMP-exosome RNA decay machinery and genomic instability [41]. However, it is not known whether the TRAMP complexes are needed to maintain TNR stability. Also, although it is known that the TRAMP and THO complexes cooperate in controlling snoRNA expression [42], cross-talk between THO and TRAMP in regulating genetic instability has not been investigated.

In the course of screening for factors that protect against fragility of an expanded CAG tract in *Saccharomyces cerevisiae*, we identified members of both the THO and TRAMP complexes. We found that CAG repeat fragility and instability increase dramatically in the absence of either Thp2 or Mft1 of the THO complex or Trf4 or Rrp6 of the TRAMP/exosome machinery. Our data show that the THO and TRAMP complexes act in complementary pathways to reduce breakage at expanded CAG repeats in a manner distinct from accumulation of stable R-loops that occur in the absence of both RNase H1 and H2. Rather, defects in either complex result in RNAPII accumulation at the repeat tract and increased genome-wide transcription-replication conflicts (TRCs). Despite the similar phenotypes, genetic analysis indicates that the initial defect that leads to TRCs likely differs for THO and TRAMP mutants. THO mutants have a more marked accumulation of RNAPII that can be released by RNase H1 overexpression to reduce chromosome fragility. In contrast, defects in the TRAMP4 complex lead to reduced RPA availability, which is responsible for some of the TRCs and chromosome breaks. Our results highlight the importance of RNA export and processing factors in stabilizing expanded TNRs and preventing chromosome fragility at structure-forming repeats.

## Results

### The THO and TRAMP complexes both protect against chromosome breakage and repeat contractions

To study the role of RNA biogenesis and surveillance mechanisms on breakage and instability of an expanded CAG repeat tract, we used a previously established yeast artificial chromosome (YAC) system containing 70 CAG repeats (CAG-70) to evaluate the rate of fragility and frequency of repeat instability (Fig 1A, [43]). When DNA double-strand breaks (DSBs) occur at the CAG repeats, repair can occur by single-strand annealing or end joining to re-ligate the broken CAG repeat, causing contractions [44]. Alternatively, if repair fails or significant resection to the $(G_4T_4)_{13}$ telomere seed occurs, this can result in de novo synthesis of a new telomere and YAC end loss (Fig 1A). This event results in loss of the *URA3* gene, which confers resistance to 5-fluoroorotic acid (5-FOA). The rate of FOA resistance (FOA$^R$), as a proxy for CAG repeat fragility, was measured by growing cells with a known starting repeat tract length for 6 to 7 generations without selection to allow fragility to occur, and then calculating the proportion of daughter yeast colonies that can grow on FOA media compared to non-selective media. Transcription occurs through the expanded CAG repeat tract on this YAC, with the

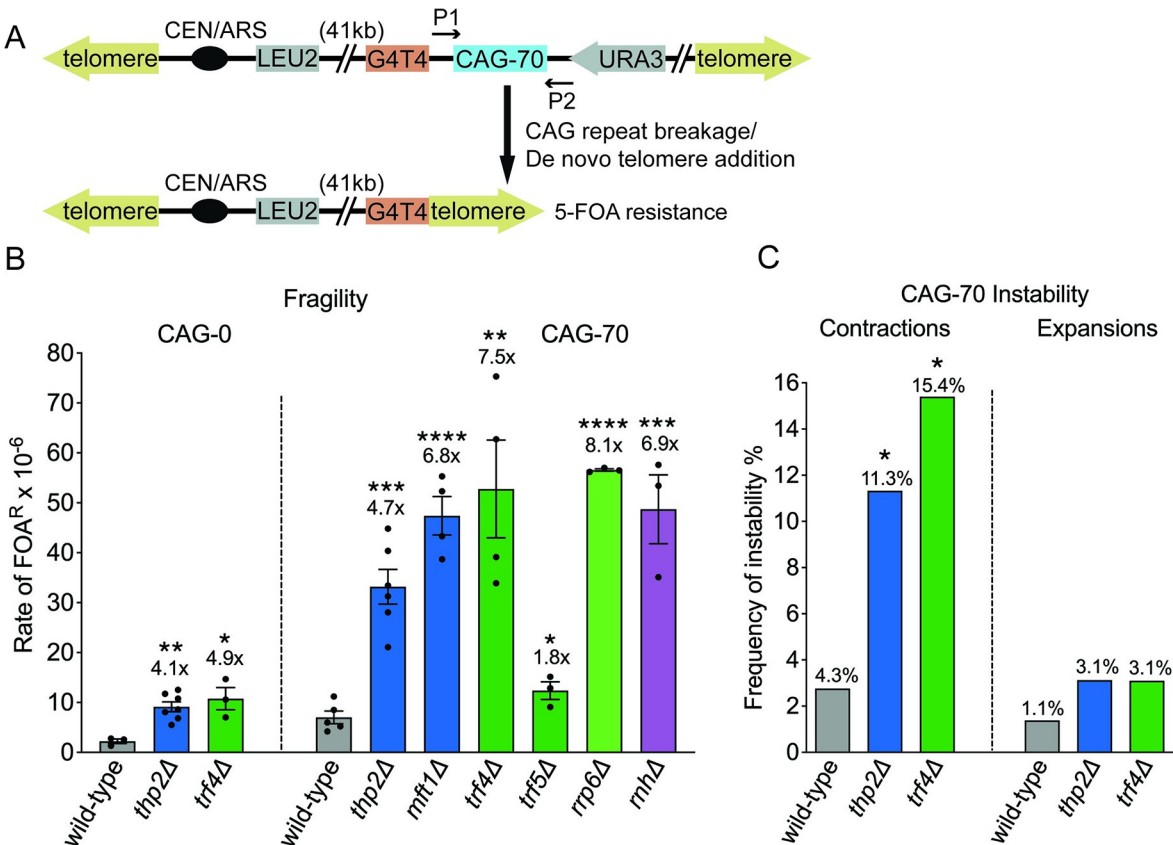

**Fig 1. CAG repeat fragility increases in mutant strains defective for THO, TRAMP4, and exosome complex members.** (A) Assay system for CAG fragility. P1, P2 with arrows indicate the site-specific primers used for sizing CAG repeats by PCR amplification. (B) Rate of FOA$^R$ × 10$^{-6}$ in indicated mutants; THO genes, blue bars; TRAMP or exosome genes, green bars. $\Delta$, gene knock-out by replacing the target gene ORF with a selectable marker; each data point represents an individual biological replicate of a 10-colony assay. Each assay was performed using 10 independent colonies and the rate determined by the method of maximum likelihood. $^*p < 0.05$, $^{**}p < 0.01$ and $^{***}p < 0.001$, $^{****}p < 0.0001$ compared to wild type with same CAG tract, by $t$ test. Average of at least 3 replicates ± SEM (standard error of the mean) is shown (Tables A and C in S1 Tables). $rnh\Delta$: $rnh1\Delta rnh201\Delta$, data from [7]. (C) CAG repeat instability is increased in the $thp2\Delta$ or $trf4\Delta$ mutants. CAG-70 contraction and expansion frequencies in indicated mutants, a minimum of 130 PCR reactions per mutant; $^*p < 0.05$, compared to wild type, Fisher's exact test (Table D in S1 Tables).

majority of transcripts emanating from readthrough transcription from the *URA3* gene (Fig 1A), which generates an rCUG transcript [7,12].

We initially identified a *thp2Δ* mutant as a top hit in a screen for yeast gene deletions that increase fragility of a CAG-85 repeat tract (see supplement of [45] for screen details). To confirm this phenotype, we deleted Thp2, a subunit of the THO complex, in a different strain background with a YAC carrying 70 CAG repeats and found that the *thp2Δ* strain exhibited a significant 4.7-fold increase in the rate of fragility compared to the wild-type strain (Fig 1B; Table A in S1 Tables). A CAG-0 control also shows a significant 4-fold increase in fragility in *thp2Δ* compared to the wild-type strain (Fig 1B; Table C in S1 Tables), indicating that the *thp2Δ* fragility phenotype is not unique to expanded triplet repeats. Contraction frequency of CAG repeats in the *thp2Δ* mutant also increases significantly, 2.6-fold over wild type (Fig 1C; Table D in S1 Tables). To confirm that FOA resistance was due to YAC end loss and not to other mechanisms such as point mutations occurring at the *URA3* marker gene, the structure of the YAC in multiple independently derived FOA-resistant colonies was examined by Southern blot (method described in [43]). The results showed a 100% end-loss frequency with all the

YAC structures consistent with de novo telomere addition at the $G_4T_4$ seed sequence (Table E in S1 Tables), indicating that loss of *URA3* is the cause of increased FOA resistance in the *thp2Δ* mutant. To confirm the importance of the THO complex in preventing CAG repeat breakage, another THO subunit, Mft1, was deleted. We chose Mft1 as it was identified in a second iteration of the screen for CAG fragility. The *mft1Δ* strain also exhibited a significant increase in CAG-70 repeat fragility that was similar to the level observed in the *thp2Δ* mutant (Fig 1B; Table A in S1 Tables). Hpr1 and Tho2 deletions confer a strong growth defect [46] and were therefore not tested. The similar *thp2Δ* and *mft1Δ* phenotype confirmed that a defective THO complex causes an increase in fragility of expanded CAG repeats.

In human genomes, transcription through expanded CAG repeat loci can occur bidirectionally to generate structured rCAG and rCUG repeat transcripts whose accumulation can be toxic to cells [47]. Such excess CAG repeat-containing RNA could be targeted for degradation by the TRAMP complex and nuclear exosome. Mutants in Trf4 of TRAMP, the 3′–5′ riboexonuclease component of the nuclear exosome Rrp6, and Lrp1 that forms a heterodimer with Rrp6 and regulates its exonucleolytic activity [48] were all found to be strong positives in subsequent iterations of the CAG-85 fragility screen (Tufts University Molecular Genetics Project Lab course 2015, 2017). To investigate whether this RNA surveillance mechanism has an impact on CAG repeat fragility, we deleted Trf4, a non-canonical poly(A)-polymerase of the TRAMP4 complex, in the YAC-containing strains. In the *trf4Δ* mutant, the rate of CAG-70 repeat fragility is significantly and dramatically elevated 7.5-fold compared to the wild-type control (Fig 1B; Table A in S1 Tables), and the CAG repeat contraction frequency is also significantly increased by 3.6-fold compared to the wild type (Fig 1C; Table D in S1 Tables). Loss of *URA3* was confirmed by PCR (Table E in S1 Tables), indicating that FOA resistance is due to CAG fragility causing loss of the right arm of the YAC. A significant elevation was also seen in the *trf4Δ* no-tract control with a 4.9-fold increase over the wild-type no-tract control (Fig 1B; Table C in S1 Tables), indicating that the TRAMP complex also protects against chromosome fragility at non-repetitive DNA. Notably, we did not see such a dramatic increase in fragility of the CAG repeat when we knocked out the other TRAMP poly(A)-polymerase, Trf5. Although the increase in CAG-70 fragility in the *trf5Δ* mutant is significant compared to the wild-type control, it is only elevated by 1.8-fold compared to 7.5-fold for the *trf4Δ* mutant (Fig 1B; Table A in S1 Tables). In previous studies, Trf4 and Trf5 were shown to have functionally distinct roles in polyadenylation of different RNA species with only partial redundancy [49]. We infer that most excess transcripts of expanded CAG repeats are polyadenylated and targeted for exosome degradation by the TRAMP4 complex containing Trf4. Deletion of Rrp6 of the exosome also led to a significantly elevated rate of CAG-70 repeat fragility compared to the wild-type control (Fig 1B; Table A in S1 Tables). We conclude that a functional TRAMP4-mediated RNA degradation mechanism protects expanded CAG repeats against fragility and deletions.

In either *thp2Δ* or *trf4Δ* mutants, CAG repeat expansion frequency was only mildly increased in contrast to the significantly and dramatically elevated contraction frequency (Fig 1C; Table D in S1 Tables). Therefore, there is a strong bias to contractions in THO and TRAMP mutants. Past research has shown that expansions often occur during gap repair pathways, whereas DSB repair within expanded CAG tracts most often lead to contractions [50,51]; therefore, the bias to contractions is consistent with the high levels of breakage caused by deletion of Thp2 or Trf4.

## THO and TRAMP4 complexes cooperatively prevent fragility of expanded CAG repeats

In order to test if THO and TRAMP4 work in the same or different pathways in preventing CAG repeat fragility, we made a *thp2Δtrf4Δ* double mutant containing the CAG-70 YAC.

These double mutants had a severe growth defect on normal yeast complete (YC) medium, compared to their isogenic wild-type control and *thp2Δ* or *trf4Δ* single mutants: viability of *thp2Δtrf4Δ* double mutants in YC-Leu growth media to maintain the YAC is reduced to only 10%, almost an 8-fold decrease compared to the wild-type control (S1A Fig; Table F in S1 Tables). These results suggest that the THO and TRAMP complexes have complementary functions in processing RNA and that both are crucial for cell growth and fitness. Due to the low cell viability of the *thp2Δtrf4Δ* mutant, we were able to obtain only a few colonies carrying full-length CAG-70 tracts; therefore, it was not feasible to perform our regular fragility proto-col, which utilizes 10 single colonies per assay, with this double mutant. Therefore, we carried out a revised 1-colony fragility assay as described in Materials and methods. Similar control assays were also performed with the wild type, *thp2Δ*, and *trf4Δ* strains. By using this modified fragility assay, we found that the frequency of FOA resistance drastically increases in the *thp2Δtrf4Δ* mutant, around 20,000-fold higher compared to the wild type, and around 10,000-fold higher compared to the *thp2Δ* and *trf4Δ* single mutants (Fig 2A; Table B in S1 Tables; significance of $p < 0.0001$ compared to *thp2Δ* and $p = 0.0005$ compared to wild type and *trf4Δ*). Such a high frequency of 5-FOA resistance in the double mutant provides evidence of a large amount of DNA breakage at the repeats in the *thp2Δtrf4Δ* mutant and explains the low viability. The synergism between Thp2 and Trf4 in preventing CAG fragility indicates that THO and TRAMP4 act in complementary pathways to prevent breakage or repair breaks.

## R-loops play a partial role in causing CAG repeat fragility in a THO mutant but no detectable role in a TRAMP4-defective background

In previous studies, abolishment of either the THO or TRAMP complex has been shown to increase R-loop-associated genome instability [22,36,37,52]. Recently, the THO complex was shown to prevent R-loop formation throughout the cell cycle, whereas other factors such as Senataxin only resolve R-loops during S-phase [53]. We previously showed that R-loops are detected at expanded CAG tracts in vivo and that increasing R-loop accumulation by remov-ing both RNase H1 and RNase H2 proteins using an *rnh1Δrnh201Δ* double mutant further increased CAG repeat fragility [7]. Interestingly, the rate of CAG-70 repeat fragility in either *thp2Δ* or *trf4Δ* is very similar to the rate of the *rnh1Δrnh201Δ* (*rnhΔ*) mutant (Fig 1B; Table A in S1 Tables). In order to test if CAG repeat fragility is caused by an R-loop-mediated mecha-nism in the *thp2Δ* or *trf4Δ* mutants, we created the *thp2Δrnh1Δrnh201Δ* and *trf4Δrnh1Δrnh201Δ* triple deletion strains. We found that CAG fragility in both *thp2ΔrnhΔ* and *trf4ΔrnhΔ* was significantly and synergistically elevated compared to their single mutants (Fig 2B; Table A in S1 Tables) and viability was decreased (S1A Fig; Table F in S1 Tables). The fragility for *thp2ΔrnhΔ* shows a 40-fold increase over the wild-type control and an 8.4-fold increase over *thp2Δ*, while the fragility for *trf4ΔrnhΔ* is even higher with a 74-fold increase over the wild-type control and a 10.5-fold increase over *trf4Δ* (Fig 2B; Table A in S1 Tables). These data demonstrate that accumulation of R-loops that occurs in the absence of RNase H processing causes CAG repeat fragility through a different pathway than RNA biogenesis defects due to defective THO or TRAMP4 complexes.

In order to test if R-loops physically accumulate at CAG repeats in either *thp2Δ* or *trf4Δ* mutants, we performed DNA:RNA immunoprecipitation coupled with qPCR (DRIP-qPCR) by using primers flanking the CAG repeats (Fig 3A, Table P in S1 Tables; Materials and meth-ods). We were not able to detect a significant increase in R-loops at the CAG repeat in either *thp2Δ* or *trf4Δ* mutants (Figs 3B and S2A and S2B; Table G in S1 Tables), unlike the 2-fold increase detected at the CAG-70 tract in the *rnh1Δrnh201Δ* mutant. RNase H treatment of the samples showed a decreased signal, confirming that the S9.6 antibody was detecting RNA:

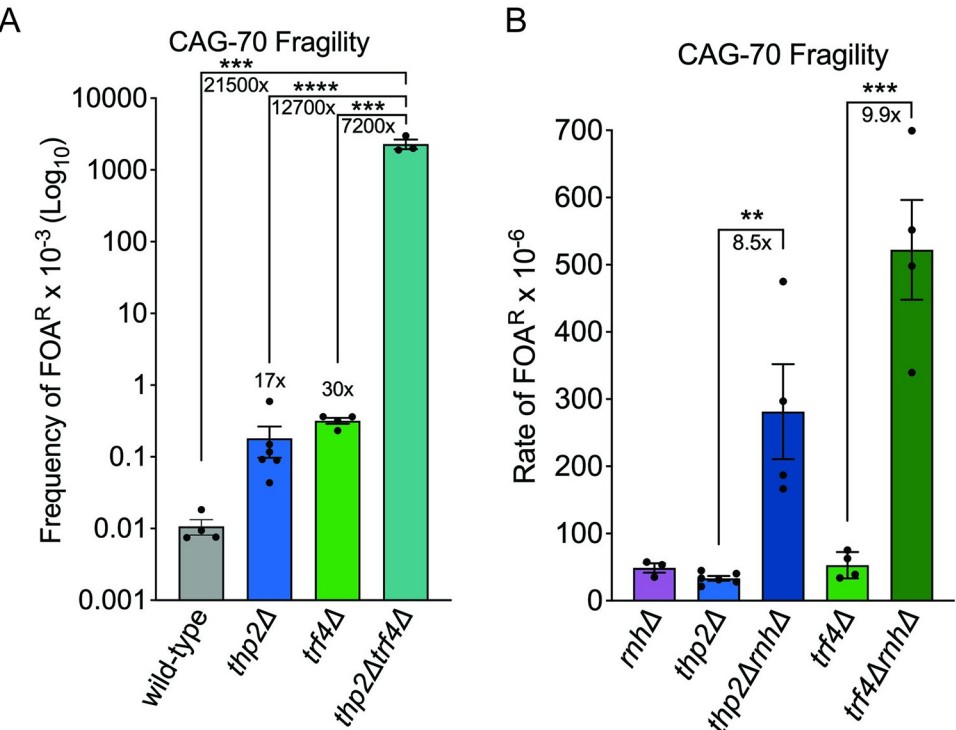

**Fig 2. Loss of Thp2, Trf4, and RNase H synergistically increase CAG repeat fragility.** (A) Frequency of FOA$^R$ × $10^{-3}$ in indicated mutants; each data point represents an individual biological replicate (for this set of experiments only, a 1-colony assay was performed instead of a 10-colony assay; see Materials and methods); $***p < 0.001$, $****p < 0.0001$, by $t$ test. Average of at least 3 experiments ± SEM is shown (Table B in S1 Tables). (B) Rate of FOA$^R$ × $10^{-6}$ in indicated mutants; $rnh\Delta$: $rnh1\Delta rnh201\Delta$, data from [7]; each data point represents an individual biological replicate of a 10-colony assay; $**p < 0.01$, and $***p < 0.001$ compared to $thp2\Delta$ or $trf4\Delta$ single mutants, by $t$ test. Average of at least 3 experiments ± SEM is shown (Table A in S1 Tables).

DNA hybrids. Though we note that only about half of the signal in the $trf4\Delta$ mutant was RNase H sensitive, indicating that the antibody may be detecting some double-stranded RNA in this background [54–56]. Detection of RNA-DNA hybrids was also confirmed at the *PMA1* locus in the $rnh1\Delta rnh201\Delta$ mutant as a positive control [27] but levels were not significantly elevated in either $thp2\Delta$ or $trf4\Delta$ mutants (S2A Fig; Table G in S1 Tables). Readthrough transcription through the CAG tract was confirmed to be similar to wild-type levels in both $thp2\Delta$ or $trf4\Delta$ mutants (Fig 3C; Table H in S1 Tables). Therefore, we conclude that if R-loops form at the CAG tract in these mutants, they are of a type not easily detected by DRIP.

Interestingly, the strand-specific transcription analysis of the CAG tract revealed that there is a 2-fold increase in cryptic antisense transcription in the $trf4\Delta$ mutant (Fig 3C; Table H in S1 Tables). In wild-type cells, there is 5.4-fold more rCUG (resulting from *URA3* readthrough) than rCAG (resulting from antisense transcription) transcript, whereas this ratio is 3.3-fold in $trf4\Delta$ mutants. This is in line with previous studies that detected excess cryptic transcripts in this mutant [34,57].

To test if R-loop formation influences CAG repeat fragility in the RNA biogenesis mutants using a functional assay, we inserted an inducible *MET25* promoter [58] before the *RNH1* gene, which encodes RNase H1, to allow induced overexpression of *RNH1* in vivo in synthetic medium lacking methionine. We confirmed overexpression of RNase H1 by using a reverse-transcription reaction and quantitative PCR (qRT-PCR) in the wild-type background as well as $thp2\Delta$ and $trf4\Delta$ mutants (S1B Fig, Table I in S1 Tables). Fragility analysis showed a 40%

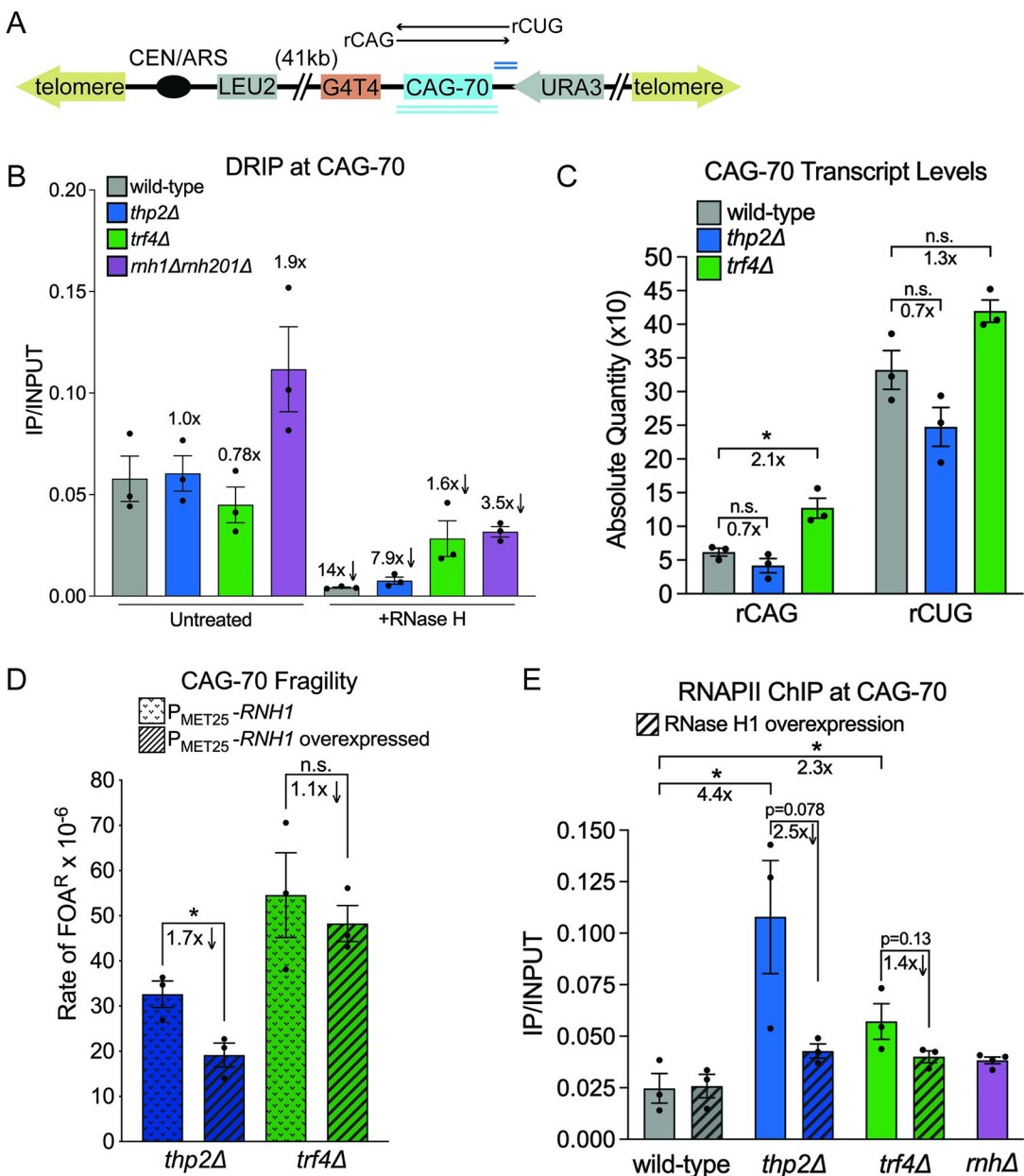

**Fig 3. Analysis of R-loop presence and RNA polymerase stalling and effect on CAG-70 fragility in RNA biogenesis mutants.** (A) The CAG-70 repeat locus analyzed. Double underlines indicate the qPCR amplicons used for qPCR: light blue is across CAG, dark blue is CAG adjacent; see S2B Fig for a comparison. (B) DRIP using the S9.6 antibody to RNA:DNA hybrids in wild-type, *thp2Δ*, *trf4Δ*, and *rnh1Δrnh201Δ* strains with and without RNase H treatment. The presence of the CAG locus was determined by qPCR using the across-CAG primers. Each bar represents the mean ± SEM of at least 3 biological replicates; each data point represents an individual biological replicate (Table G in S1 Tables). (C) qRT-PCR detecting rCUG and rCAG transcript levels. A consistent amount RNA per sample was reverse transcribed into cDNA by strand-specific RT-PCR and qPCR was used to quantify cDNA levels at the indicated locus. rCUG transcripts result from readthrough transcription from the *URA3* gene [12], rCAG transcripts result from cryptic transcription from the G4T4 locus direction. Each bar represents the mean ± SEM of at least 3 biological replicates; each data point represents an individual biological replicate; $^*p < 0.05$, compared to the same transcript in wild-type cells, by $t$ test (Table H in S1 Tables). (D) Rate of FOA$^R$ × $10^{-6}$ in indicated mutants in RNase H1 overexpression conditions. The P$_{MET25}$-*RNH1* strain was grown in the presence of methionine (*RNH1* repressed) or absence of methionine (*RNH1* induced). Each data point represents an individual biological replicate; $^*p < 0.05$, compared to no induction condition in the same mutant, by $t$ test. Average of at least 3 experiments ± SEM is shown (Table A in S1 Tables). See S1B Fig and Table I in S1 Tables for RNase H1 expression levels in the various strains. (E) RNAPII ChIP at the CAG-70 repeat locus in the indicated strains either without (*RNH1* endogenous promoter) or with RNase H1 overexpression (*RNH1* expressed under the P$_{MET25}$ promoter, induced in the absence of methionine). The IP/INPUT signal at the CAG repeat was determined by

qPCR using the primer set adjacent to CAG-70. Each bar represents the mean ± SEM of at least 3 biological replicates; each data point represents an individual biological replicate; *$p < 0.05$ compared to wild type compared to no RNase H1 overexpression, by $t$ test (Table J in S1 Tables). ChIP, chromatin immunoprecipitation; DRIP, DNA:RNA immunoprecipitation; FOA, fluoroorotic acid; RNAPII, RNA polymerase II.

reduction (1.7-fold decrease) in the rate of FOA resistance compared to the un-induced condition for the *thp2Δ* mutant ($p = 0.027$; Fig 3D and S1C Fig; Table A in S1 Tables). These results indicate that depletion of Thp2 causes a functional accumulation R-loops at the CAG repeats, consistent with previous findings that R-loops accumulate in THO mutants [22,27,52,59]. We conclude that R-loops within the CAG tract in the THO-defective mutant are different than those that accumulate in the *rnh1Δrnh201Δ* mutant and may be shorter or more transient since they were not detected by DRIP [60] but do contribute to causing the increase in CAG fragility observed in this background.

In contrast, no reduction in the FOA$^R$ rate was seen in the *trf4Δ* background upon RNase H1 overexpression (Figs 3D and S1C; Table A in S1 Tables). Combined with the failure to detect increased R-loops at the CAG tract by DRIP in the *trf4Δ* mutant, these results suggest that the increased CAG fragility in this background is not primarily a result of R-loop accumulation but rather due to another mechanism.

## Loss of either THO or TRAMP4 results in RNA polymerase II accumulation at expanded CAG repeats, but for different reasons

The largest RNA polymerase II complex (RNAPII) subunit Rpb1 is targeted for degradation in the absence of THO subunit Tho2, suggesting RNAPII stalling occurs in THO-defective mutants [61]. To explore whether RNAPII was stalled at expanded CAG repeats in either *thp2Δ* or *trf4Δ* mutants, we performed RNAPII chromatin immunoprecipitation (ChIP) analysis. Indeed, we discovered that RNAPII was significantly enriched at the CAG repeat tract compared to wild type by about 4.4-fold in the *thp2Δ* mutant and 2.3-fold in the *trf4Δ* mutant (Figs 3E and S2C; Table J in S1 Tables; $p = 0.043$ *thp2Δ* to wild type, $p = 0.044$ *trf4Δ* to wild type). In contrast, we did not detect a significant increase in RNAPII enrichment at CAG-70 repeat tract in the *rnh1Δrnh201Δ* mutant (Figs 3E and S2C; Table J in S1 Tables; $p = 0.084$). A similar pattern was observed at 2 gene loci, the highly transcribed *ACT1* locus and the lower transcribed *MMR1* locus, with 5- to 9-fold increases in RNAPII occupancy in *thp2Δ* and 5- to 8-fold increase in *trf4Δ* strains, indicating that RNAPII accumulation on chromatin may be universally increased in these mutants (S2C Fig).

There is a possibility that RNAPII enrichment may be due to either increased transcription through a genomic region or RNAPII stalling. Since there is no significant increase in rCUG/rCAG transcript levels in the *thp2Δ* mutant compared to wild type (Fig 3C; Table H in S1 Tables), RNAPII enrichment is likely due to RNAPII stalling in that mutant. However, we cannot rule out a transcription effect in the *trf4Δ* mutant, as there is a 2.1-fold increase in rCAG and 1.3-fold increase in rCUG transcripts compared to wild type (Fig 3C; Table H in S1 Tables), though this could also be due to accumulation of unprocessed transcripts rather than increased transcription.

To better understand the reason for RNAPII accumulation at the CAG tract in the THO and TRAMP mutants, we tested how RNase H1 overexpression affects RNAPII levels at the CAG repeat tract using strains containing the inducible P$_{MET25}$-*RNH1* gene. In wild-type cells, no change in RNAPII levels at the CAG tract was detected upon endogenous *RNH1* overexpression. Interestingly, overexpression of RNase H1 reduced the amount of RNAPII at the CAG tract in the *thp2Δ* strain by 2.5-fold to almost wild-type levels (Fig 3E; Table J in S1 Tables). These data suggest that digestion of hybrids is releasing RNAPII stalled in the *thp2Δ* mutant.

In contrast, RNase H1 overexpression resulted in only a mild suppression of RNAPII enrichment at CAG repeats in the *trf4Δ* mutant (1.4-fold decrease, Fig 3E; Table J in S1 Tables), consistent with the inability of RNase H1 overexpression to suppress CAG fragility. These results suggest that the cause of RNAPII accumulation in TRAMP mutants is not due to an RNAPII elongation or dissociation defect and may be indirect.

## Loss of either THO or TRAMP4 results in increased transcription-replication conflicts (TRCs) genome-wide

One possible explanation for DSBs at the expanded CAG tract is TRCs due to a stalled RNAPII complex interfering with DNA replication [62]. To determine whether TRCs were occurring in *thp2Δ*, *trf4Δ*, or *rnh1Δrnh201Δ* mutants, we employed a proximity ligation assay (PLA) using antibodies against PCNA to detect the replisome and RNAPII to detect the transcription machinery. This assay detects proteins that are within 40 nm of each other [63], which are detectable as foci in the nucleus (example nuclei are shown in Fig 4A). The size of the foci is comparable to what has been observed for this assay in mammalian nuclei, though they appear comparatively larger due to the small size of the yeast nucleus (see scale bars in Fig 4A and in [64,65] as examples). Some cells had foci detected outside of the nucleus, but the percentage of these compared to total foci counted within the nucleus was small and similar across strains (Table K in S1 Tables). Mitotic arrest with nocodazole treatment resulted in elimination of almost all foci, consistent with expectations of an S-phase event (S4A Fig; Table L in S1 Tables).

Quantification of the number of foci in each strain showed a highly significant increase between wild type and each indicated mutant, with the effect being smallest for the *rnh1Δrnh201Δ* strain (1.3-fold increase, $p < 0.001$), followed by a further significant 1.6-fold increase in the *thp2Δ* nuclei, and an even more striking 2.2-fold increase in *trf4Δ* nuclei ($p < 0.0001$ for both) (Fig 4B; Table L in S1 Tables). These data indicate that all 3 mutants cause TRCs genome-wide; however, the effect is significantly stronger in the THO and TRAMP mutants compared to the RNase H mutant strain. Interestingly, the PLA data correlate more closely with the levels of RNAPII detected by ChIP within the CAG-70 tract and less well with the levels of R-loops within the repeat tract detected by DRIP. These data suggest that a stalled RNAPII, with or without an accompanying R-loop, is a stronger barrier to DNA replication than R-loops that persist in the absence of RNase H processing.

To further understand the basis of the TRCs, we analyzed numbers of PLA foci in the strains with endogenously inducible RNase H1. Overexpression of RNase H1 reduced PLA foci numbers in both *thp2Δ* and *trf4Δ* to wild-type levels, (Fig 4C; Table L in S1 Tables). There was no significant reduction in PLA foci upon RNase H1 overexpression in a wild-type strain (Fig 4C; Table L in S1 Tables). These data suggest that RNase H1 resolves RNA:DNA hybrids that form at the sites of conflicts between RNAPII and a replication fork in THO and TRAMP4 mutants. Considering the effect of RNase H1 overexpression on both PLA foci and RNAPII ChIP, we propose that overexpression of RNase H1 degrades RNAPII-associated hybrids in the THO mutant, to release stalled RNAPII and resolve TRCs. However, since neither R-loops nor clear evidence for RNAPII release were detected in TRAMP mutants, the hybrids causing TRCs in this background are likely of a different nature (see Discussion).

## Cells lacking Trf4 exhibit an RPA availability deficiency that leads to CAG repeat fragility and TRCs

Since RNase H1 overexpression did not rescue CAG fragility in the *trf4Δ* mutant, we investigated other possible reasons for the increased breaks that occur in the absence of TRAMP4. RPA recruitment is decreased in the absence of Trf4 or Rrp6 after resection of DSBs, and this

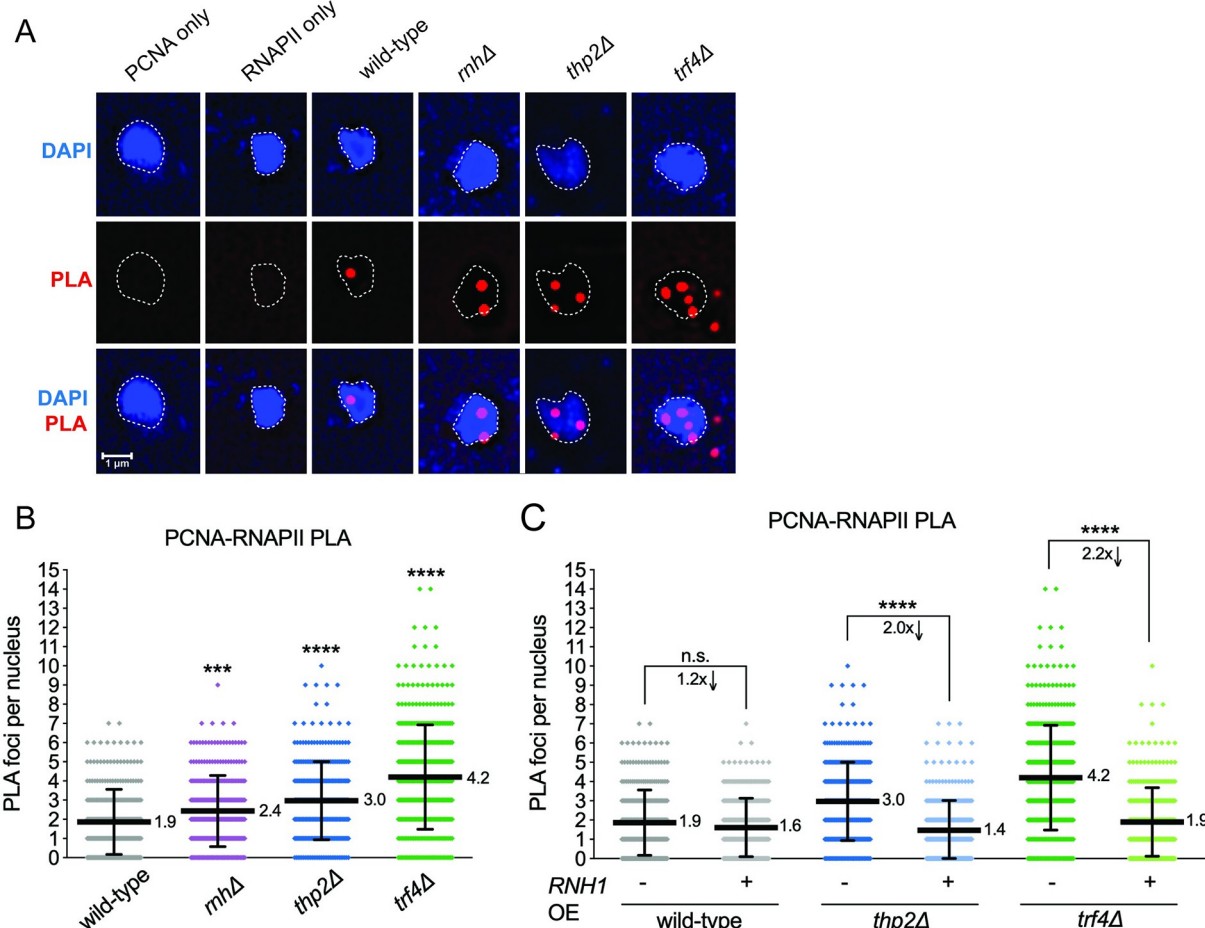

**Fig 4. TRCs in RNA biogenesis mutants.** (A) Example images for the PLA. DAPI staining the nucleus is shown in blue, PLA foci (proximity of PCNA and RNAPII-pSer2) shown in red. The first 2 columns are single-antibody control conditions and the other columns with both antibodies. (B) PLA using an antibody to RNAPII-pSer2 and one to PCNA to assess TRCs genome-wide in the indicated strains. $N \geq 300$ nuclei quantified per condition. At least 3 experiments were performed for each strain or condition with 100 nuclei screened per biological replicate. Data points indicate individual PLA foci counts per nucleus (see S2 Tables for raw PLA foci counts). Horizontal bars with adjacent numbers indicate the mean number of foci. Error bars show mean ± SD. ***$p < 0.001$ or ****$p < 0.0001$ compared to wild type by Mann–Whitney test (Table L in S1 Tables). (C) PLA assessing TRCs upon RNase H1 overexpression. PLA was performed in the indicated strains either without (*RNH1* endogenous promoter) or with RNase H1 overexpression (*RNH1* expressed under the $P_{MET25}$ promoter, induced in the absence of methionine). Experiment was performed and analyzed as stated in panel B. PLA, proximity ligation assay; RNAPII, RNA polymerase II; TRC, transcription-replication conflict.

attenuates the Mec1/ATR response to DSBs as well as to fork-stalling agents HU and MMS [66]. *trf4Δ* does not affect resection of an HO-induced break [66]. Additionally, the catalytic RNA degradation activity of the mammalian Rrp6 homolog EXOSC10 is needed for normal levels of RPA recruitment to sites of irradiation-induced DSBs [67]. RPA-loading is expected to be crucial for preventing hairpin formation at single-stranded CAG or CTG repeats [68], and indeed, RPA was shown to accumulate at CAG repeats by induction of convergent transcription [69]. To test if RPA recruitment was impaired when RNA degradation factors were deleted, we performed ChIP to detect levels of RPA at the CAG repeat using an RPA antibody that recognizes all 3 yeast RPA subunits. We observed a 3-fold increase in enrichment of RPA proteins at the CAG-70 repeat over a locus within the *ACT1* gene (*ACT1* locus) (Figs 5A and S1), indicating that more RPA is recruited to the expanded CAG repeat tract than a regular genomic locus. The RPA ChIP signal at the CAG-70 repeats in the *trf4Δ* strain was reduced

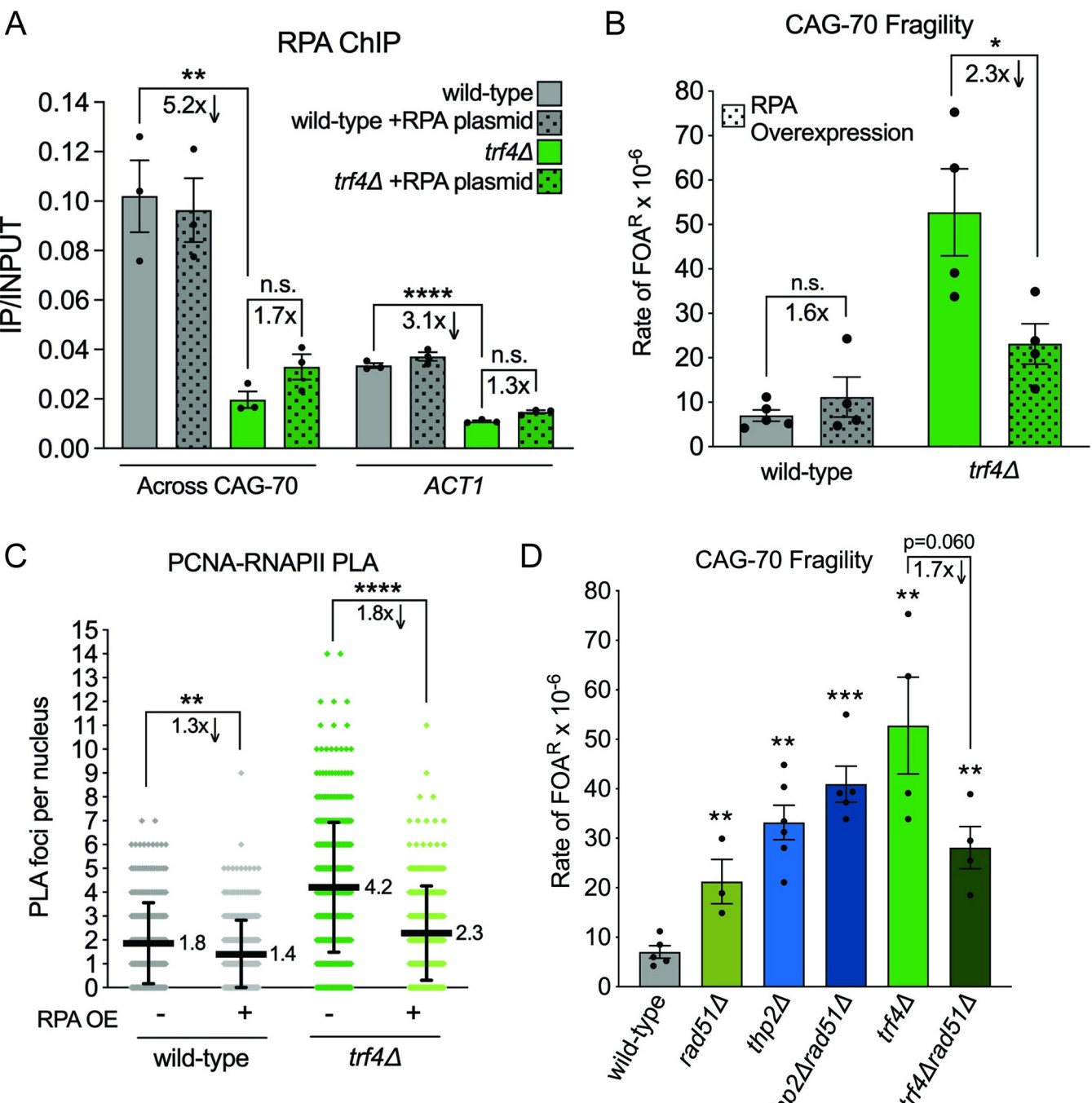

**Fig 5. RPA-loading deficiency and genetic interactions in the *trf4Δ* strain.** (A) RPA ChIP at the CAG repeat or *ACT1* locus using an antibody to RPA (detects Rfa1, Rfa2, and Rfa3) in the indicated strains, shown as the IP/INPUT signal at the CAG-70. RPA overexpression was achieved by expression of *RFA1*, *RFA2*, and *RFA3* from a 2μ multicopy plasmid (strains transformed with RPA overexpression vector compared to no vector; see S3 Fig). Each bar represents the mean ± SEM of at least 3 biological replicates and each data point represents an individual biological replicate; ** $p < 0.01$ and **** $p < 0.0001$, compared to wild type by *t* test (Table M in S1 Tables). (B) Rate of FOA$^R$ × 10$^{-6}$ in indicated mutants and RPA overexpression conditions (strains transformed with RPA overexpression vector compared to no vector). Each data point represents an individual biological replicate; * $p < 0.05$, compared to no induction condition in the same mutant, by *t* test. Average of at least 3 experiments ± SEM is shown (Table A in S1 Tables). (C) PLA assessing TRCs in strains with and without RPA overexpression (strains transformed with RPA overexpression vector compared to no vector). Experiment was performed and analyzed as stated in panel 4B. ** $p < 0.01$ and **** $p < 0.0001$ compared to indicated strain by Mann–Whitney test (Table L in S1 Tables). (D) Rate of FOA$^R$ × 10$^{-6}$ in indicated mutants; each data point represents an individual fragility experiment (biological replicate); ** $p < 0.01$ and *** $p < 0.001$, compared to wild type with same CAG tract, ^$p = 0.06$, compared to *trf4Δ* by *t* test. Average of at least 3 experiments ± SEM is shown; each data point is 1 biological replicate (Table A in S1 Tables). ChIP, chromatin immunoprecipitation; FOA, fluoroorotic acid; PLA, proximity ligation assay; RNAPII, RNA polymerase II.

significantly by 5.2-fold compared to the wild-type strain ($p = 0.0053$) (Figs 5A and S3C; Table M in S1 Tables). These data indicate that when lacking Trf4, RPA association at the CAG-70 repeat is diminished. RPA levels were also reduced by 3.1-fold at the *ACT1* locus in *trf4Δ* cells (Figs 5A and S3C), indicating that there is a global deficiency in RPA association with DNA in this background.

A reduced level of RPA binding is predicted to increase the chance of hairpin formation in exposed ssDNA regions, which could interfere with DNA replication or cause inefficient repair, increasing the chance of breakage or failure to repair, resulting in YAC end loss. To test whether the reduced RPA binding observed at the CAG tract in the *trf4Δ* strain was contributing to the increase in fragility, we overexpressed RPA by introducing a multicopy plasmid containing all 3 RFA genes [70]. This led to a 20- to 40-fold increase in *RFA1*, *RFA2*, and *RFA3* expression over basal levels, and a 3- to 20-fold increase in protein levels detected by western blot (S3A Fig; Tables N and O in S1 Tables). We observed a significant suppression of CAG tract fragility upon overexpression of RPA in the *trf4Δ* strain (44% reduction, 2.3-fold decrease) that was not observed in wild-type cells (Fig 5B; Table A in S1 Tables). This result indicates that the reduced RPA binding observed at the CAG tract in the *trf4Δ* strain is a source of increased repeat tract fragility in the absence of TRAMP4 activity. To further understand how RPA overexpression is suppressing CAG fragility in the *trf4Δ* strain, we examined if RPA overexpression would restore RPA protein levels at the CAG repeat. We saw a mild increase in RPA detected by ChIP at the CAG repeat in the *trf4Δ* strain, though this was not statistically significant (Figs 5A and S3C; Table M in S1 Tables). It is possible that even a mild restoration of RPA at the CAG repeat tract upon RPA overexpression can reduce CAG repeat fragility, or another mechanism is responsible for suppression of DNA breakage.

To determine whether reduced RPA binding to DNA was having a genome-wide consequence when the TRAMP4 complex is defective, we tested whether RPA overexpression could relieve the increased level of TRCs observed. Indeed, TRCs in the *trf4Δ* strain were suppressed to wild-type levels upon RPA overexpression (1.8-fold decrease, $p < 0.0001$; Fig 5C; Table L in S1 Tables). This result indicates that lack of RPA availability leads to the increased TRCs observed in the absence of TRAMP4. Though less dramatic, TRCs were also reduced upon RPA overexpression in wild-type cells (1.3-fold, $p = 0.001$; Fig 5C; Table L in S1 Tables), indicating that RPA may also be limiting at spontaneous TRCs that occur, impacting their resolution.

Finally, to determine whether recombinational repair or fork protection was required to recover from the TRCs observed in RNA biogenesis mutants and prevent fragility, we tested the role of the Rad51 protein. CAG-70 fragility levels were similar for *thp2Δ* and *thp2Δrad51Δ* cells, indicating that the damage occurring at the CAG tract in THO mutants is generally not rescued by homologous recombination (Fig 5D; Table A in S1 Tables). Surprisingly, fragility was reduced in the *trf4Δrad51Δ* mutant (Fig 5D; Table A in S1 Tables). Therefore, the presence of Rad51 is exacerbating CAG fragility in cells lacking TRAMP4. In situations where forks are de-protected or cannot restart, excess Rad51 binding can lead to deleterious accumulation of recombination structures at stalled forks [71]. Therefore, the suppressive effect of deleting Rad51 in the *trf4Δ* strain may be due to RPA depletion at CAG stalled forks that allows for excessive Rad51 binding and formation of deleterious recombination intermediates, leading to increased fragility.

## Discussion

In this study, we discovered that CAG repeat fragility is greatly increased in the absence of components of the THO or TRAMP complexes or the exosome, indicating that defects in

RNA biogenesis and processing lead to frequent occurrence of chromosome breakage events. We examined several potential causes of the fragility in the *thp2Δ* (THO) and *trf4Δ* (TRAMP4) mutants, including R-loop accumulation, RNA polymerase stalling, TRCs, and defects in RPA recruitment. We found that RNAPII stalling at expanded CAG repeats was the most prominent phenotype for *thp2Δ* mutant and is likely the main initiator of the increased CAG fragility in THO mutants. We also found that genome-wide TRCs were significantly increased in both *thp2Δ* and *trf4Δ* mutants, to a much greater degree than a strain missing both RNase H1 and RNase H2 that has a confirmed increase in RNA:DNA hybrids. Consistent with our in vivo data, a recent in vitro study using *Escherichia coli* proteins showed that RNA polymerase transcription complexes, especially if oriented head-on with replication, created stable blockages that were more severe than an R-loop without an attached RNA polymerase [72]. Based on the 5.4-fold greater level of transcription in the rCUG orientation at the CAG repeat studied here, we predict that most conflicts within the CAG tract will be in the head-on orientation. In TRAMP mutants, an additional factor comes into play, and our data support that depletion of RPA levels available to bind the genome are a primary cause of TRCs that lead to chromosome fragility. Altogether, our results show that defects in both RNP formation on nascent RNA and RNA processing lead to RNAPII stalling and TRCs and that chromosome fragility is a frequent and deleterious consequence.

Since the THO-defective *hpr1Δ* mutant was shown to exhibit increased R-loop formation and TAR [22], and R-loop accumulation in RNase H mutants increases CAG repeat fragility [7], we predicted that R-loops would be the main cause of fragility at the expanded CAG tract in the THO mutant. Unexpectedly, we were not able to detect increased R-loop signals at the CAG tract in the *thp2Δ* mutant by DRIP as we did for RNase H1 and H2 depletion, though we did see a partial but significant decrease in CAG fragility when RNase H1 was overexpressed in the *thp2Δ* strain. Also, TRCs were dramatically reduced by in vivo RNase H1 overexpression. Because we could not directly detect an increase in R-loops at the CAG tract by DRIP in the *thp2Δ* mutant, but only detected their phenotypic consequence by the RNase H1 overexpression experiments, we hypothesize that they are transient and likely associated with the stalled RNAPII, rather than stable R-loops left after passage of the transcriptional machinery. It has been recently suggested that small R-loops (60 bp or less) associated with paused RNA-PII may not be well detected by DRIP, which preferentially detects longer (>200 bp), more stable R-loops [60]. It is also possible that short hybrids still associated with RNAPII are shielded from detection by the S9.6 antibody used in the DRIP protocol. Based on the role of THO in binding nascent RNA to direct it to the nuclear pore complex for export, it is probable that the unbound nascent RNA is more likely to re-anneal to the DNA template in this mutant, inhibiting RNAPII progression and causing an accumulation of RNAPII within the CAG tract (Fig 6, middle pathway). Indeed, THO is the complex that binds closest to RNAPII and has a direct role in transcription elongation [19]. Since overexpression of RNase H1 decreased RNAPII stalling at the CAG repeat tract and TRCs genome-wide in the *thp2Δ* mutant, these tethered RNA polymerases likely block the replication fork, leading to TRCs, fragility, and repeat contractions (Fig 6, middle). Our result that RNAPII stalling is suppressed upon RNase H1 overexpression suggests that digestion of the RNA of a co-transcriptional RNA:DNA hybrid could release stalled RNAPII, leading to the observed suppression of TRCs and breaks. Consistent with our results, a recent study observed increased RNAPII stalling in a Senataxin mutant (*sen1-3*) at the rDNA replication fork barrier (rRFB) [73]. Stalling at the rRFB in *sen1-3* was suppressed upon overexpression of human RNase H1 [73]. Also similar to our model, another recent study proposed that in *sen1* mutants, the main replication barrier is a RNAPII complex trapped on the DNA template due to stable R-loop formation [74]. In contrast, in an *rnh1Δrnh201Δ* mutant, the major contributor to CAG repeat fragility and instability is through

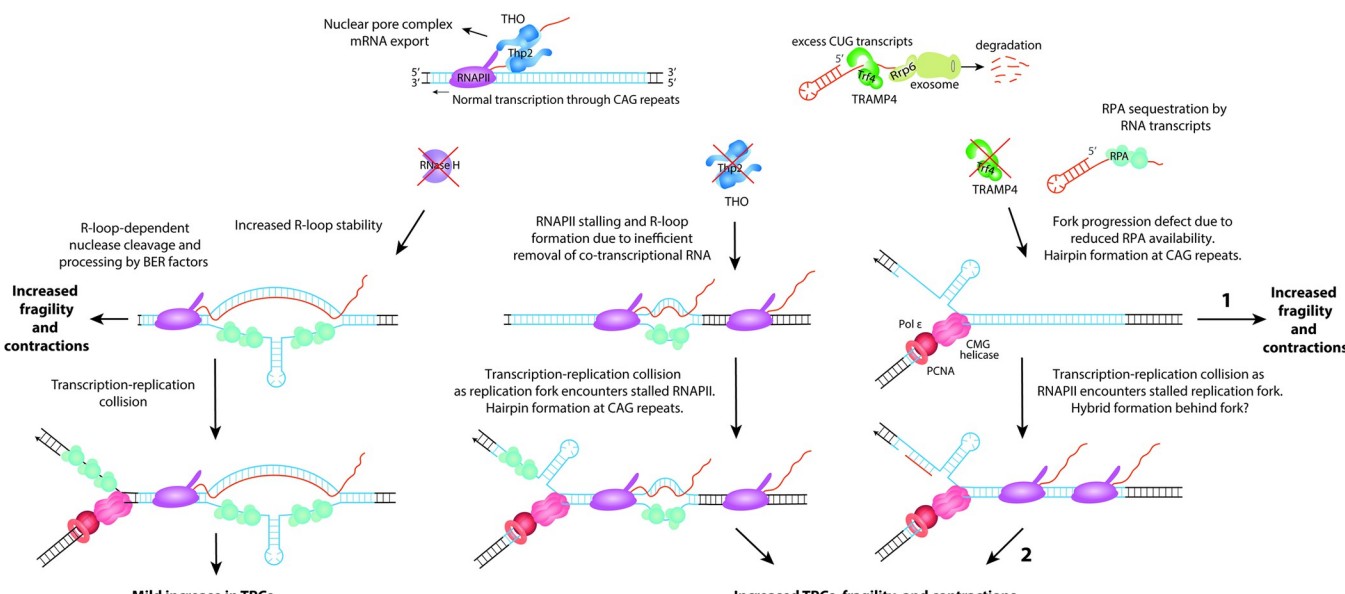

**Fig 6. Model for CAG repeat fragility arising from defective THO and TRAMP4 complexes.** Transcription through CAG repeats is shown (top), in the direction of the readthrough transcription from the *URA3* gene on the YAC (rCUG transcript), though some cryptic transcription occurs from the other direction as well (rCAG transcript). Under wild-type conditions, RNAPII generates rCAG or rCUG transcripts, which are co-transcriptionally bound by THO (blue). They may be recognized as CUTs, bound and polyadenylated by TRAMP4 (green), and targeted for exosome-mediated degradation through Rrp6 and the exosome (olive green). In the absence of RNase H1 and H2 (left), there is an accumulation of R-loops, including at the CAG repeat tract. These R-loops are targeted by the MutLγ nuclease and processed by BER factors, resulting in increased CAG repeat fragility and contractions (see [7]). There is also a mild increase in TRCs in the absence of RNase H1 and H2. In the absence of THO (middle), RNAPII stalling and R-loop formation occur due to inefficient removal of nascent RNA that hybridizes to the exposed ssDNA. In this scenario, RNAPII stalling is the initiating event. The increased RNAPII stalling leads to TRCs as the replication fork approaches and encounters stalled RNAPII. The stalled replication fork and exposed ssDNA allows for CAG hairpin formation, leading to fragility and contractions of the CAG repeats. In the absence of TRAMP4 (right), a replication fork progression defect occurs due to impaired RPA loading, allowing for hairpin formation at the CAG repeats and replication fork stalling, leading to fragility and contractions (TRAMP4 side, fragility pathway 1). In this scenario, replication fork stalling is the initiating event. TRCs occur as RNAPII encounters the stalled replication fork, potentially accompanied by short RNA: DNA hybrid formation. This ultimately leads to additional fragility and contractions of the CAG repeats (TRAMP4 side, fragility pathway 2). The balance between fragility pathway 1 and pathway 2 could be different at a natural site of fork stalling, like the CAG-70 tract, compared to highly transcribed gene bodies. CUT, cryptic unstable transcript; RNAPII, RNA polymerase II; ssDNA, single-stranded DNA; TRC, transcription-replication conflict; YAC, yeast artificial chromosome.

R-loop-mediated mechanisms, and we did not detect increased RNAPII stalling at the CAG tract in this condition (Fig 6, left) [7]. Therefore, both types of transcription-dependent events can lead to TRCs and genome instability, but they differ in their characteristics and do not necessarily co-occur.

Our results with the Thp2 mutant highlight the importance of successful transcription elongation in preventing TRCs and chromosome breaks not only within genes, but also in repetitive DNA. Mutation of other proteins involved in mRNA maturation also has genome-protective effects. A mutation in Ysh1, an endoribonuclease subunit of the mRNA cleavage and polyadenylation complex, resulted in slowed RNAPII passage through a GAA repeat tract, and GAA expansions and fragility in an R-loop-independent manner [75]. Excess Yra1, a member of the mRNA export TREX complex, causes genome-wide replication slowing and increased DNA damage, ultimately causing telomere shortening and replicative senescence [76,77]. Thus, interference at multiple points of the RNA maturation process can lead to similar outcomes. A recent study showed that human THOC7, a component of human THO, accumulates in transcriptionally active repeat regions and defects are associated with γH2AX foci, suggesting that our results will be relevant to maintenance of repeats in the human genome [78].

The TRAMP complexes target mainly noncoding RNAs to Rrp6 and the nuclear exosome for degradation, a pathway that is highly conserved from yeast to humans. It was previously shown that *trf4Δ* mutants exhibit increased transcription-dependent recombination and an elevated mutation rate that was linked to the presence of excess nascent RNA transcripts [36]. These genome instability phenotypes were attributed to increased R-loops since they were reduced upon overexpression of RNase H1 [36]. Cryptic telomeric RNA and TERRA are also increased in *trf4Δ* mutants [34,57]. Strains depleted of Trf4 or Rrp6 showed increased chromosome loss and terminal deletions that were suppressed upon RNase H1 expression [37] and were suggested to be caused by in-trans R-loops that are dependent on Rad51 and Rad52 [40]. However, at the CAG tract, we could find no evidence of increased R-loops by DRIP or suppression of fragility by RNase H1 overexpression. Instead, we found a high level of genome-wide TRCs in the *trf4Δ* mutant. Therefore, we explored whether the primary cause of CAG fragility and TRCs in the absence of a functional TRAMP4 complex could be related to RPA levels on DNA. Based on our observation of a significant decrease in RPA binding to both CAG repeats and the *ACT1* gene and a suppression of both CAG repeat fragility and TRCs upon RPA overexpression in the *trf4Δ* mutants, we propose an alternative hypothesis for genome instability phenotypes in the absence of TRAMP4 (Fig 6, right). A deficiency of RPA binding to replication forks paused within the CAG tract could allow more hairpin formation or could expose other naturally stalled forks to excess degradation, either of which could result in decreased fork recovery. This would lead to increased chromosome fragility, as well as repeat contractions (Fig 6, TRAMP4 side, fragility pathway 1). A secondary consequence would be RNAPII colliding with the stalled forks to increase TRCs and RNA:DNA hybrid formation (either at the stalled fork or stalled RNAPII), which would further inhibit recovery from the TRC (Fig 6, TRAMP4 side, fragility pathway 2). RNaseH1 overexpression could remove these stalled fork-associated hybrids to resolve TRCs. This model is consistent with the observed suppression in CAG repeat fragility and genome-wide TRCs upon overexpression of RPA in the *trf4Δ* mutant. Our model can also explain the suppression of fragility by loss of Rad51, since a fork unprotected by RPA binding will be more available for Rad51 loading and unregulated HR, which could lead to cleavage or failure to effectively restart. At sites of R-loop-mediated TRCs, RECQ5 is needed to disrupt RAD51 filaments to allow for fork restart that is mediated by MUS81 cleavage [79]. Nonetheless, other models are possible, and it will be interesting to further investigate the events that lead to genome instability in TRAMP and exosome mutants.

The question remains of why RPA levels on DNA are reduced in TRAMP4 mutants. TRAMP mutants have an excess accumulation of cryptic RNA [32,34,36,42], and we also observed a 2-fold increase in cryptic rCAG RNA at the CAG tract (Fig 3C). A recent study showed that RPA can bind with high affinity to ssRNA [80]. Therefore, RPA loading at stalled forks could be decreased due to excess unprocessed RNAs in the nucleoplasm of *trf4Δ* cells (both rCUG/rCAG transcripts and other transcripts) competing for RPA binding. Another consideration is that RNA:DNA hybrids appear to contribute to the TRCs observed in the *trf4Δ* mutant as we observed a suppression of TRCs upon overexpression of RNase H1, indicating that degradation of hybrids allows for TRC resolution. It has been recently proposed that hybrids form at stalled forks and interfere with fork restart [81,82]. These RNA:DNA hybrids can result from either de novo synthesis by RNAPII or from the post-replicative hybridization of nascent RNA after a fork passes the transcription machinery (reviewed in [83,84]). We propose that low levels of RPA due to sequestration by excess RNAs in the absence of TRAMP4 favors formation of hybrids behind the fork, inhibiting TRC resolution (Fig 6, right). It is also possible that the TRAMP complex and exosome are themselves able to degrade RNA within hybrids and without these complexes the hybrids block RPA binding.

The exosome is important for preventing TRCs in human cells [85], and in EXOSC10 (yeast Rrp6)-depleted cells, RPA recruitment to a DSB was restored upon clearance of damage-induced long noncoding RNAs (dilncRNAs) by treatment with RNase H1 [67]. Altogether, our data support that the TRAMP4 complex plays an important role in preventing TRCs and chromosomal breaks through processing of RNAs, preventing their accumulation and either sequestration or blocking of RPA.

Even though the THO and TRAMP complexes are both involved in RNA biogenesis, we found that deletion of both pathways was highly synergistic for CAG fragility. In addition, deletion of Trf4 along with RNase H1 and RNase H2 was strongly synergistic, consistent with our conclusion that *trf4* mutants cause CAG fragility by a pathway not limited to increasing R-loops. The *thp2ΔrnhΔ* triple mutant also showed synergistic fragility, though less than the other combinations, in line with THO mutants causing CAG fragility through both R-loop and non-R-loop mechanisms. Our results indicate that RNA biogenesis mutants can cause genome instability by multiple mechanisms that are only partially overlapping, including interference with RNAPII progression or unloading, TRCs, R-loops, and reduced RPA binding to DNA. Since the fragility phenotypes were not just additive but synergistic, it implies that the activities of these protein complexes work in a cooperative manner to prevent genome instability. The conversion of TRCs to a DSB could occur by several mechanisms and the downstream events that lead to chromosome fragility will be an interesting area of future investigation.

In summary, our results highlight the importance of multiple aspects of RNA biogenesis in preventing genome stability and show that these mechanisms are especially crucial at structure-forming repeats. Since repetitive DNA occurs throughout genomes, these pathways are expected to be of paramount importance in preventing breaks and the ensuing deleterious consequences.

## Materials and methods

### Yeast strains and genetic manipulation

Yeast strains used in this study are listed in Table Q in S1 Tables. Yeast knock-out mutants were created by one-step gene replacement [86] using selectable markers, *KANMX6*, *TRP1*, or *HIS3MX6* and method described in [86]. In each case, at least 2 independent strains were created and verified as correct gene replacements lacking the replaced ORF. The *MET25* promoter was inserted directly upstream of the *RNH1* gene by homology-directed replacement; the construct of the *MET25* promoter with a *natNT2* marker gene is from pYM-N35 [58]. Overexpression of RPA was achieved by transforming desired strains with a 2μ multicopy plasmid containing *RFA1*, *RFA2*, and *RFA3* [70]. PCR was used to confirm the successful knock-out of a gene by confirming the presence of the selectable marker and the absence of the endogenous gene at the target locus. CAG tract length was verified in all successful clones by PCR as described below.

### CAG repeat tract length verification and instability analysis

The CAG tract was verified by using PCR amplification of genomic DNA from yeast colonies using colony PCR with primers specific to the CAG repeat tract listed in Table P in S1 Tables. The PCR protocol was described in [44] and the product sizes were analyzed by high-resolution gel electrophoresis. For instability analysis, colonies with a verified (CAG)70 tract length were grown in YC-Ura-Leu liquid media and plated for singles in YC-Ura-Leu plates. A total of 24 colonies per starting culture (this is 1 biological replicate) were screened for CAG tract

length by PCR until a total of at least 130 colonies per strain was screened. See also [87,88] for a detailed protocol.

### Fragility analysis by YAC end loss

A total of 10 independent colonies carrying the correct length CAG tract were used in 1 fragility assay as described in [44], referred to as a 10-colony fragility assay. See also [87,88] for a detailed protocol. The colonies grown on FOA-Leu and YC-Leu were counted, and the rate of FOA resistance was calculated by using the Ma-Sandri-Sarkar Maximum Likelihood Estimator (MSS-MLE) [89,90]. At least 3 independent 10-colony fragility assays were performed for each condition, which would represent 30 independent colonies assayed at minimum. At least 2 independently created strains were tested for each mutant, except for strains containing P$_{MET25}$-*RNH1* for RNase H1 overexpression experiments (only 1 strain was created per mutant). The loss of the *URA3* marker on the YAC for at least 10 independent FOA-resistant colonies was confirmed by either Southern blot or PCR for the wild-type, *thp2Δ*, and *trf4Δ* mutants. Due to high FOA resistance and repeat contraction frequencies in the *thp2Δtrf4Δ* mutant, a regular 10-colony fragility assay and the MSS-Maximum Likelihood estimation were not applicable. Therefore, a 1-colony fragility assay was carried out for this strain (Fig 2A; Table B in S1 Tables). Each assay was from 1 parental colony with desired CAG-70 tract length. The mutant cells were grown in YC-Leu media for an extended time (approximately 48 hours) to achieve approximately 5 doublings instead of 6 to 7 divisions in regular assays. Then, the FOA-resistance frequency was calculated by counting the cells grown on the FOA-Leu and YC-Leu plates.

### DNA:RNA immunoprecipitation (DRIP)

DRIP was performed by using the same procedure as described in [91] with the following changes. Locus-specific restriction sites were chosen to fragment the target sites prior to DNA: RNA hybrid pulldown using S9.6 antibody. Restriction enzymes used for fragmenting DNA around loci of interest were pooled and applied to the lysed samples (HindIII+BsaI for CAG-70, AflII+HaeII for *MMR1*, and BglII for *PMA1*). Per each IP sample, 2 μL of each enzyme were used to digest the target site. After genomic DNA was digested with restriction enzymes, DNA was purified with 3 μL StrataClean Resin and then through Sephadex g-50 columns as described in [91]. The purified DNA was then split in half: one half was used for S9.6 (4 μg, Kerafast) antibody binding and protein A Dynabead pulldown; the other half was treated with 8 μL RNase H (NEB) overnight at 37˚C [91]. Approximately 20 μL of digested DNA from both conditions (treated and untreated with RNase H) were taken as INPUT DNA samples, prior to addition of S9.6 antibody and pulldown [91]. After the DNA:RNA hybrids were eluted from the beads and S9.6 antibody, samples were treated with Proteinase K. The eluted DNA products (both INPUT and IP samples) were purified using NucleoSpin Gel and PCR Clean-up Kit (Takara). The purified DNA products were quantified by using SYBR Premix Ex Taq II (Tli RNase H Plus) (Takara Bio) on a LightCycler 480 II (Roche). qPCR reactions were performed in technical duplicate, and the average is shown. Each experiment was done with 3 biological replicates. All the primers used for qPCR are listed in Table P in S1 Tables.

### Chromatin immunoprecipitation (ChIP)

RPA and RNAPII ChIP were done using the same procedure as the ChIP described in [7], with the following changes. At least 3 biological replicates were done for each condition. Cells were grown to log-phase (around 0.6 to 0.8 OD) in YC-Ura-Leu, YC-Met-Cys-Ura-Leu (for RNase H1 overexpression), or YC-His-Ura-Leu (for RPA overexpression). P$_{GAL1}$-CAG-100

strains (for RNAPII ChIP) were grown in YC-Ura-Leu + 2% raffinose for 3 to 4 doublings until reaching at OD approximately 1; cells were washed and split into glucose and galactose-containing YC-Ura-Leu media and grown for 1 hour. Unsynchronized cells were cross-linked in 1% formaldehyde for 20 minutes at room temperature. DNA was sheared by sonication. Approximately 40 μL of Protein G Dynabeads (Invitrogen) was used for the immunoprecipitation (IP) for RPA (Figs 5A and S3C); 30 uL of ChIP-Grade Protein G Magnetic Beads (Cell Signaling Technology) for RNAPII ChIP (Figs 3E and S2C). Antibody usage is as follows: anti-RPA (Agrisera AS07 214; 5 uL of 1:4 dilution from original stock for each sample (concentration undetermined by the company due to its serum format); anti-RNAPII (BioLegend (ordered VWR 10019–922) RBP1 [8WG16]; 5 μg per sample). IP and input DNA levels were quantified by qPCR (performed in technical duplicate with the average shown) with SYBR green PCR mastermix (Roche), SYBR Premix Ex Taq II (Tli RNase H Plus) (Takara Bio), or 2X Universal SYBR Green Fast qPCR (ABclonal) on a LightCycler 480 II (Roche) or QuantStudio 6 real-time PCR system (Applied Biosystems).

## Quantitative reverse transcription PCR (qRT-PCR)

Yeast strains were grown to an OD of approximately 0.6 to 1 for collection. Three biological replicates were used for each set of experiments (except 2 replicates for *RNH1* levels, S1B Fig). Cells were treated with 50 to 100 U of zymolyase in a 1 M sorbitol/100 mM EDTA solution. Cells were lysed and RNA was extracted using the RNeasy kit and RNase-free DNase set (QIAGEN). cDNA was prepared using the Superscript First Strand Synthesis kit (Life Technologies); oligo d(T) primers or locus specific primers (see Table P in S1 Tables) were used for priming during reverse transcription. RT-PCR samples were analyzed using qPCR (performed in technical duplicate with the average shown) with SYBR Premix Ex Taq II (Tli RNase H Plus) (Takara Bio), Power SYBR Green PCR Master Mix (Applied Biosystems), or 2X Universal SYBR Green Fast qPCR (ABclonal) on a QuantStudio 6 real-time PCR system (Applied Biosystems).

## Trichloroacetic acid (TCA) protein precipitation and western blotting

Whole-cell extracts were prepared TCA protein precipitation was performed as previously described [92]. Three biological replicates from either 2 (for strains containing no vector) or 3 isolated strains (strains transformed with RPA overexpression vector) were used. RPA protein levels were analyzed by SDS-PAGE and western blotting using antibody to RPA (Agrisera AS07-214; 1:5,000). Blots were imaged using chemiluminescent detection (Pierce ECL Western Blotting Substrate, Thermo Scientific) followed by imaging on Bio-Rad ChemiDoc XRS + Molecular Imager. Rfa1, Rfa2, and Rfa3 protein levels were quantified relative to wild type using BioRad Image Lab (using relative volume tool). Expression was normalized to total protein (detected by Ponceau S staining). For total protein detection, blots were stained with Ponceau S for 15-minute post-transfer and imaged on Bio-Rad ChemiDoc XRS+ Molecular Imager. See S1 Raw images for all raw western blot images used for quantification.

## Proximity ligation assay

PLAs were performed using the Duolink kit from Millipore Sigma. Preparation of yeast cells was adapted from [93]. Cells were grown in YC-Ura-Leu media, YC-Ura-Leu-Met-Cys media for RNase H1 overexpression, YC-His-Ura-Leu for RPA overexpression PLA experiments, and in YPD with a 1-hour nocodazole treatment (15 ug/mL concentration) for nocodazole experiments (to suppress TRCs) at 30°C to log phase. Cells were fixed with 4% paraformaldehyde for 15 minutes at room temperature. Cells were washed 3× with wash buffer (1.2 M

sorbitol in 100 mM $KPO_4$ (pH 6.5)). Cell walls were digested in zymolyase solution (500 μg/mL Zymolyase 100T, 1.2 M sorbitol, 100 mM $KPO_4$ (pH 6.5), 20 mM 2-Mercaptoethanol) at 30˚C shaking for 30 minutes. Cells were washed 3× with wash buffer and resuspended in 1.2 M sorbitol. Cells were attached to poly-L-lysine-coated slides and washed 3× with wash buffer and 3× with permeabilizing solution (1% TritonX-100 in 100 mM $KPO_4$ (pH 6.5)). Cells were blocked with the provided blocking solution for 30 minutes at 37˚C (Duolink kit). Cells were incubated with primary antibody (1:400 RNAPII [pSer2] Novus Biologicals NB100-1805, 1:400 PCNA [5E6/2] abcam ab70472) overnight at 4˚C. Cells were washed 2× in wash buffer A (Duolink kit) and incubated in anti-rabbit PLUS and anti-mouse MINUS probes diluted 1:5 in antibody diluent (Duolink kit) for 1 hour at 37˚C. Cells were washed 2× in wash buffer A and incubated in ligase solution (1:40 ligase in 5× ligase buffer diluted 1:5 in water, Duolink kit) for 30 minutes at 37˚C. Cells were washed 2× in wash buffer A and incubated in amplification solution (1:80 polymerase in 5× amplification buffer diluted 1:5 in water, Duolink kit) for 100 minutes at 37˚C. Cells were washed 2× in wash buffer B (Duolink kit) and 1× in 0.01× wash buffer B. The coverslip was mounted with In Situ Mounting Medium with DAPI (Duolink kit) and sealed with nail polish. Slides were imaged using a Leica Dmi8 Thunder or a DeltaVision Ultra microscope at 100× oiled objective. The number of PLA foci per DAPI-stained nucleus was quantified by manual counting. The assay was done with 3 biological replicates (2 for the nocodazole experiment) from either 1 ($P_{MET25}$-RNH1 strains) or 2 (all other strains tested) independently created strains. At least 100 nuclei were quantified per replicate for a total of at least 300 nuclei quantified for each strain/condition (except for nocodazole experiments in which 200 nuclei were quantified).

## Supporting information

**S1 Fig. Cell viability in mutant backgrounds, *RNH1* transcript levels, and CAG fragility in RNase H1 overexpression strains.** (A) Frequencies of viability are shown. Viability is calculated by comparing the amount of cells that grew into colonies on YC-Leu plates to the amount of cells plated, counted by hemacytometer. Each data point represents an individual biological replicate; average ± SEM is shown; $^*p < 0.05$, compared to wild-type, *t* test (Table F in S1 Tables). (B) *RNH1* transcript levels in 3 conditions: the native *RNH1* gene under its own promoter, the *RNH1* gene placed under the *MET25* promoter ($P_{MET25}$-RNH1) uninduced, and $P_{MET25}$-RNH1 induced in media lacking methionine and cysteine. mRNA was reversed transcribed into cDNA by RT-PCR and qPCR was used to quantify cDNA at *RNH1* and *ACT1* gene loci. *RNH1* qPCR signal was normalized to the *ACT1* qPCR signal in the indicated mutants (Table I in S1 Tables). (C) Rate of $FOA^R \times 10^{-6}$ in indicated strains containing $P_{MET25}$-RNH1 grown with methionine (no Rnh1 induction) or without methionine (Rnh1 induced); each data point represents an individual biological replicate of a 10-colony assay; $^*p < 0.05$, compared to no induction condition in the same mutant, by *t* test. Average of at least 3 experiments ± SEM is shown (Table A in S1 Tables). Note that even under non-induced conditions (+ methionine), the *RNH1* gene is slightly overexpressed when under control of the non-native *MET25* promoter and the fragility rates are higher than when *RNH1* is expressed under its native promoter, especially for the wild-type strain (compare rate to Fig 1B). (TIF)

**S2 Fig. DRIP-qPCR at CAG-70, *URA3*, *MMR1*, *PMA1* loci and RNAPII ChIP at CAG-70, ACT1, and MMR1 loci.** (A) DRIP-qPCR shown as IP/INPUT in wild-type, *thp2Δ*, t*rf4Δ*, and *rnh1Δrnh201Δ* strains with and without RNase H treatment. Each bar represents the mean ± SEM of at least 3 biological replicates; each data point represents an individual biological replicate. *PMA1* and *MMR1* loci were evaluated as loci previously shown to have high and

low R-loops, respectively [27,94], as a comparison to the CAG locus. $^*p < 0.05$ compared to wild-type, $^\wedge p < 0.05$ and $^{\wedge\wedge}p < 0.01$ compared to corresponding strain with no treatment, by $t$ test (Table G in S1 Tables). (B) DRIP-qPCR comparing results using primer sets that either span the CAG-70 tract (across-CAG, 345 bp) or are directly adjacent to the CAG-70 tract (CAG adjacent, 101 bp; see Fig 3A) in wild-type, $thp2\Delta$, $trf4\Delta$, and $rnh1\Delta rnh201\Delta$ strains with and without RNase H treatment. Each bar represents the mean ± SEM of at least 2 biological replicates; each data point represents an individual biological replicate. $^*p < 0.05$ compared to wild-type, $^\wedge p < 0.05$, $^{\wedge\wedge}p < 0.01$, and $^{\wedge\wedge\wedge\wedge}p < 0.0001$ compared to corresponding strain with no treatment, by $t$ test (Table G in S1 Tables). (C) RNAPII ChIP shown as IP/INPUT in the indicated strains either without (*RNH1* endogenous promoter) or with RNase H1 overexpression (*RNH1* expressed under the $P_{MET25}$ promoter, induced in the absence of methionine). A wild-type strain with a galactose inducible promoter driving transcription through a CAG-100 repeat tract was done as a positive control (cells are grown in raffinose-containing media and then split into glucose and galactose-containing media; growth in glucose is no induction, growth in galactose is induction of transcription). The *OLI1* locus was included as a negative control since *OLI1* is in the mitochondrial genome and is not transcribed by RNAPII. Each bar represents the mean ± SEM of at least 3 biological replicates; each data point represents an individual biological replicate. $^*p < 0.05$, $^{**}p < 0.01$, $^{***}p < 0.001$ compared to wild-type or strain indicated by bracket (comparing glucose to galactose condition), by $t$ test (Table J in S1 Tables).
(TIF)

**S3 Fig. RPA transcript and protein levels and RPA ChIP upon RPA overexpression.** (A) RPA overexpression was achieved by expression of *RFA1*, *RFA2*, and *RFA3* from 2μ multicopy plasmid (strains transformed with *RFA1/RFA2/RFA3* 2μ multicopy plasmid compared to no vector). mRNA was reversed transcribed into cDNA by RT-PCR and qPCR was used to quantify cDNA at *RFA1*, *RFA2*, *RFA3* and *ACT1* gene loci. *RFA1/RFA2/RFA3* were normalized to the *ACT1* qPCR signal in the indicated strains (Table N in S1 Tables). (B) Rfa1, Rfa2, and Rfa3 protein levels quantified in wild-type and *trf4Δ* strains with and without RPA overexpression (strains transformed with *RFA1/RFA2/RFA3* 2μ multicopy plasmid compared to no vector). Each data point represents the quantification of Rfa1/Rfa2/Rfa3 protein levels relative to wild type (Table O in S1 Tables). Three separate biological replicates were analyzed for protein quantity. A representative western blot either hybridized to the RPA antibody (detecting Rfa1, Rfa2, and Rfa3) or stained for total protein levels with Ponceau S are shown below the graph. See S1 Raw images for all raw western blot images used for quantification. Different exposures of blots were used to quantify Rfa1, Rfa2, and Rfa3 protein levels (shown is 1 example blot with the exposure used to quantify Rfa3). (C) RPA ChIP shown as IP/INPUT in wild-type and t*rf4Δ* strains with and without RPA overexpression. Use of an RPA antibody for immunoprecipitation and IgG negative controls are shown. Each bar represents the mean ± SEM of at least 3 biological replicates; each data point represents an individual biological replicate. $^*p < 0.05$, $^{**}p < 0.01$, and $^{****}p < 0.0001$ compared to wild-type, $^{\wedge\wedge}p < 0.01$ and $^{\wedge\wedge\wedge\wedge}p < 0.0001$ compared to corresponding strain with RPA antibody used for IP, by $t$ test (Table M in S1 Tables).
(TIF)

**S4 Fig. PLA single antibody controls.** Antibodies to the Ser2 phosphorylated form of RNAPII (RNAPII-pSer2) and one to PCNA were used to assess TRCs genome-wide in the indicated strains. $N \geq 200$ (200 for nuclei for nocodazole experiments, 300 for all other experiments) quantified per condition with 100 nuclei screened per replicate. Data points indicate individual PLA foci counts per nucleus (see S2 Tables for raw PLA foci counts). Error bars show mean ± SD. (A) PLA in wild-type, *thp2Δ*, and *trf4Δ* strains with nocodazole treatment to

reduce TRCs. ****$p < 0.0001$ comparing no treatment and nocodazole treatment, by Mann–Whitney test. Heavy horizontal bars with adjacent numbers indicate the mean number of foci. (B) PLA in wild-type, *rnh1Δrnh201Δ*, *thp2Δ*, and *trf4Δ* strains double antibody (RNAPII and PCNA) experiments are shown alongside single antibody controls. (C) PLA in wild-type, *thp2Δ*, and *trf4Δ* strains either without (*RNH1* endogenous promoter) or with RNase H1 over-expression (*RNH1* expressed under the $P_{MET25}$ promoter, induced in the absence of methionine). (D) PLA in wild-type and *trf4Δ* strains with (strains transformed with *RFA1/RFA2/RFA3* 2μ multicopy plasmid) and without (strains containing no vector) RPA overexpression. See Table K in S1 Tables for quantification of foci outside nuclei (see S2 Tables for raw PLA foci counts), which averaged 20% and was similar for all strains. See Table L in S1 Tables for *p*-values of single antibody controls compared to double antibody conditions, by Mann–Whitney test.
(TIF)

**S1 Tables. Supporting information tables. Table A.** Fragility analysis (rate of FOA-resistance) of CAG-70 repeats on the *URA3*-YAC. **Table B.** One-colony fragility analysis (frequency of FOA-resistance) of CAG-70 repeats on the *URA3*-YAC. **Table C.** Fragility analysis of no tract (CAG-0) on the *URA3*-YAC. **Table D.** Instability analysis of CAG-70 repeats on the *URA3*-YAC. **Table E.** Analysis of *URA3* presence in FOA-resistant colonies. **Table F.** Viability of yeast strains on YC-Leu. **Table G.** DRIP analysis. **Table H.** CAG-70 transcript expression data. **Table I.** *RNH1* expression data. **Table J. RNAPII c**hromatin immunoprecipitation (ChIP) analysis. **Table K.** Quantification of foci detected outside of nucleus in PLA experiments. **Table L.** Proximity-Ligation Assay (PLA) data. **Table M.** RPA ChIP analysis. **Table N.** RPA expression data. **Table O.** RPA protein expression. **Table P.** Primers used in this study. **Table Q.** Yeast Strains used in this study.
(PDF)

**S2 Tables. Raw PLA data.**
(XLSX)

**S1 Raw images. Raw western blot images.** (A) Raw image of blot shown in S3B Fig. The four lanes on the right (no X) were used for the example blot image shown in S3B Fig. All lanes were used for quantification. Hybridization of RPA antibody is shown on the left and Ponceau S staining is shown on the right. (B) Raw images of blots used for quantification of Rfa1 in S3B Fig and Table O in S1 Tables. Hybridization of RPA antibody is shown above and Ponceau S staining is shown below. (C) Raw images of blots hybridized with RPA antibody used for quantification of Rfa2 in S3B Fig and Table O in S1 Tables. Ponceau S staining shown in panel B was also used for Rfa2 quantification. (D) Raw images of blots used for quantification of Rfa3 in S3B Fig and Table O in S1 Tables. Hybridization of RPA antibody is shown above and Ponceau S staining is shown below. The following method was used to capture the images. For Rfa1/Rfa2/Rfa3 expression: Blots were imaged using the chemi setting on a Bio-Rad Chemi-Doc XRS+ Molecular Imager. Appropriate exposure was used for quantification of each of the 3 RPA subunits. For Ponceau S: Blots were imaged using the epi white setting on a Bio-Rad ChemiDoc XRS+ Molecular Imager.
(PDF)

## Acknowledgments

We thank Cailin Joyce who re-tested *THP2* in the original CAG fragility screen, as well as the students in the 2015 and 2017 Tufts Molecular Genetics Project Lab who identified *TRF4*,

*MFT1*, and *LRP1* genes as potential hits. We also thank Meaghan McGoldrick who helped with strain construction and Alexandra Khristich and Sergei Mirkin for sharing the RPA over-expression plasmid.

## Author Contributions

**Conceptualization:** Rebecca E. Brown, Xiaofeng A. Su, Catherine H. Freudenreich.

**Data curation:** Rebecca E. Brown, Xiaofeng A. Su.

**Formal analysis:** Rebecca E. Brown, Xiaofeng A. Su, Catherine H. Freudenreich.

**Funding acquisition:** Catherine H. Freudenreich.

**Investigation:** Rebecca E. Brown, Xiaofeng A. Su, Stacey Fair, Katherine Wu, Lauren Verra, Robyn Jong, Kristin Andrykovich.

**Methodology:** Catherine H. Freudenreich.

**Project administration:** Catherine H. Freudenreich.

**Resources:** Catherine H. Freudenreich.

**Supervision:** Xiaofeng A. Su, Catherine H. Freudenreich.

**Validation:** Rebecca E. Brown, Xiaofeng A. Su, Catherine H. Freudenreich.

**Visualization:** Rebecca E. Brown, Xiaofeng A. Su, Catherine H. Freudenreich.

**Writing – original draft:** Xiaofeng A. Su.

**Writing – review & editing:** Rebecca E. Brown, Xiaofeng A. Su, Catherine H. Freudenreich.

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
