## [Editor Report · Decision Letter 0]

16 Dec 2021

Dear Dr Freudenreich, 

Thank you for submitting your manuscript entitled "THO and TRAMP complexes prevent transcription-replication conflicts, DNA breaks, and CAG repeat contractions" for consideration as a Research Article by PLOS Biology.

Your manuscript has now been evaluated by the PLOS Biology editorial staff, as well as by an academic editor with relevant expertise, and I am writing to let you know that we would like to send your submission out for external peer review.

Once your full submission is complete, your paper will undergo a series of checks in preparation for peer review. Once your manuscript has passed the checks it will be sent out for review. To provide the metadata for your submission, please Login to Editorial Manager (https://www.editorialmanager.com/pbiology) within two working days, i.e. by Dec 18 2021 11:59PM.

If your manuscript has been previously reviewed at another journal, PLOS Biology is willing to work with those reviews in order to avoid re-starting the process. Submission of the previous reviews is entirely optional and our ability to use them effectively will depend on the willingness of the previous journal to confirm the content of the reports and share the reviewer identities. Please note that we reserve the right to invite additional reviewers if we consider that additional/independent reviewers are needed, although we aim to avoid this as far as possible. In our experience, working with previous reviews does save time. 

If you would like to send previous reviewer reports to us, please email me at rhodge@plos.org to let me know, including the name of the previous journal and the manuscript ID the study was given, as well as attaching a point-by-point response to reviewers that details how you have or plan to address the reviewers' concerns. 

Given the disruptions resulting from the ongoing COVID-19 pandemic, please expect some delays in the editorial process. We apologise in advance for any inconvenience caused and will do our best to minimize impact as far as possible.

Kind regards,

Richard

Richard Hodge, PhD

Associate Editor, PLOS Biology

rhodge@plos.org

PLOS

---

## [Decision Letter · Decision Letter 1]

9 Feb 2022

Dear Dr Freudenreich,

Thank you for submitting your manuscript "THO and TRAMP complexes prevent transcription-replication conflicts, DNA breaks, and CAG repeat contractions" for consideration as a Research Article at PLOS Biology. Please accept my sincere apologies for the delays that you have experienced during the peer review process. Your manuscript has been evaluated by the PLOS Biology editors, an Academic Editor with relevant expertise, and by three independent reviewers.

The reviews are attached below. You will see that the reviewers find your manuscript interesting, in particular the finding that the instability of the trf4Δ mutant is due to RPA depletion. However, they raise overlapping concerns with the reporting of replicate data and with the omission of several important control experiments in the manuscript. In addition, they note the lack of evidence for R-loop accumulation despite the RNaseH overexpression fragility phenotypes. The reviewers suggest several experiments to directly resolve this, including the use of an RNAPII ChIP assay and blocking transcription through the repeat regions.

In light of the reviews, we will not be able to accept the current version of the manuscript, but we would welcome re-submission of a much-revised version that takes into account the reviewers' comments. We cannot make any decision about publication until we have seen the revised manuscript and your response to the reviewers' comments. Your revised manuscript is also likely to be sent for further evaluation by the reviewers.

We expect to receive your revised manuscript within 3 months. Please email us (plosbiology@plos.org) if you have any questions or concerns, or would like to request an extension. At this stage, your manuscript remains formally under active consideration at our journal; please notify us by email if you do not intend to submit a revision so that we may end consideration of the manuscript at PLOS Biology.

**IMPORTANT - SUBMITTING YOUR REVISION**

*Re-submission Checklist*

*Published Peer Review*

*PLOS Data Policy*

*Blot and Gel Data Policy*

Sincerely,

Richard

Richard Hodge, PhD

Associate Editor, PLOS Biology

rhodge@plos.org

REVIEWS:

Reviewer #1: Brown, et al. describe the roles of the THO and TRAMP complexes in preventing transcription-replication conflicts (TRCs), DNA breaks, and CAG repeat fragility. When Trf4 and Thp2 are knocked out, CAG repeat fragility, RNA Pol II stalling, and TRCs increase. While increases in R-loops were not observed via DRIP, the authors observed a minor rescue of CAG fragility phenotypes with RNase H overexpression in thp2Δ mutants but not in trf4Δ mutants. The trf4Δ mutants were instead shown to exhibit reduced RPA binding to the CAG repeats, leading to the original observation that CAG fragility and TRC phenotypes in trf4Δ mutants could be rescued with RPA overexpression. Overall, this manuscript is well-written and brings welcome clarifications to the mechanisms by which RNA biogenesis factors prevent CAG repeat instability. A particularly exciting and novel aspect of this manuscript is the observation that RPA overexpression can at least partially rescue the CAG fragility phenotype. This will have exciting implications for evaluating similar RPA depletion in other models of defective RNA processing. The work is also nicely referenced and connects the importance of RNA metabolic processes (e.g., RNA export and degradation) and their role in maintaining genome stability.

General Comments

One of the surprising findings is that authors did not detect R-loop accumulation at the CAG repeats in thp2Δ mutants by DRIP-qPCR. However, they report that RNase H1 oe causes a slight but significant reduction in fragility and a larger reduction in TRC as measured by PCNA-RNAPII PLAs. In their model, the authors propose that stalled RNAPIIs may be the actual drivers of the instability in this mutant but that RNase H1 oe may alleviate in part the phenotype by releasing stalled RNAPIIs. The manuscript would be significantly strengthened if this was tested directly using RNAPII ChIP over the test region with and without RNase H1.

In that same vein, the authors claim that R-loops "functionally" accumulate in the thp2Δ mutant even though they are not physically detectable. This is an interesting pirouette... I believe that in all fairness, the authors should state that they can't rule out that RNase H1 may not function on co-transcriptional R-loops but instead may possess other functions that ameliorate TRCs. The Pasero lab, for instance, has suggested that RNase H1 may help the restart of stalled forks (ref 75, 76). This disclaimer will be all the more needed if the experiment asked above does not provide the expected results… 

The point that RNase H1 may affect replication fork restart is of course, brought to the fore by the findings in the trf4Δ mutant, where RNase H1 oe has a strong effect on TRCs but no effect on fragility. This suggests that the increased fragility in that context is not due to TRCs. In fact, this raises the possibility that the instability in that context may not be due to the transcription through the CAG repeat anymore. The authors assume that the RNA species responsible for titrating RPA originate from rCUG transcripts. I would gather instead that these transcripts represent a small proportion of the transcripts that accumulate genome-wide in this mutant. The manuscript would be significantly strengthened if this was tested directly by blocking / modulating transcription through the CAG repeats.

Specific concerns: 

There are general concerns regarding experimental design in several of the figures. For example, the number of data points in different experiments is variable and it is unclear if these are from individual experiments or biological replicates. As per the legends, some experiments appear to only be performed once instead of in biological triplicate (e.g., PLA). Finally, many of the experiments are normalized, but it is unclear what they are normalized to. The wild-type samples are often not "1" (e.g., ChIP). These concerns should be clarified moving forward.

For DRIP-qPCR/ChIP: could the authors please report the raw % input values instead of normalized values (perhaps in Supp material). This is essential in order to determine the DRIP/ChIP efficiency especially since authors claim to not be able to detect some R-loops. 

Figures (General): Many of the graphs have varying numbers of datapoints. For example, in Figure 1, Wild-type has 3, thp2Δ has 7, trf4Δ has 3, etc. The legend indicates that "each dot represents an individual data point" and "an average of the experiments ± SEM are shown." Are the individual datapoints then an example of one experiment or the combination of experiments?

Figure 2: With the revised one-colony fragility assay, are wild-type cells still able to be used as a negative control? If so, these data should include the wild-type FOAR data for both A and C.

Figure 4 - How many biological replicates of PLA were included in the study? From the Figure legend, it appears that only 1 replicate was performed.

Minor comments:

Line 283: Replace the word "massive".

Line 305-306: "We previously showed that R-loops form preferentially at expanded CAG tracts in vivo and that increasing R-loop stability by removing both RNase H1 and RNase H2". I'm not sure the word "preferentially" should be used. The previous study showed a slight increased R-loops loads at the CAG tract, this does not make it "preferential". Similarly, I do not believe that the authors measured "R-loop stability" per se (half-life) in their previous work. This should be rephrased using more neutral language. 

Line 315: the adjective "stable" is not necessary. 

Line 520-522: The suggestion that DRIP may not recognize short R-loops should not be oversold. Rewrite to "It has been recently SUGGESTED that small R-loops (60 bp or less) associated with paused RNAPII MAY not BE WELL detected by DRIP, which preferentially detects longer (>200 bp), more stable R-loops [55]. 

Reviewer #2 (David Tollervey, signs review): The authors investigate the basis of CAG repeat instability using a model system in yeast, and find that loss of the RNA BP Thp2 and the TRAMP component Trf4 synergistically enhance repeat DNA fragility. Interestingly, this is attributed to different defects in the single mutants; R-loop formation in thp2∆ and TRCs due to RPA depletion in trf4∆. 

The results are clearly presented and will be of wide interest. I would support publication with only minor changes. 

Minor points:

1: The authors might also consider the possibility that the exosome/TRAMP complexes might directly prevent and/or resolve R-loops by degradation of the RNA strand. A potential increase in RNA/DNA hybrid formation in absence of Trf4 lead to reduced binding of RPA to the single stranded DNA. TRCs in the single trf4 mutant were compensated by overexpression of RNH1.

2: The "DRIP" technique reported by the Chedin lab relied on naked DNA as starting material. The authors used antibody S9.6, but performed a variant of DRIP technology that relies on cross-linked DNA. It is possible that formaldehyde crosslinking could shield R-loops, thereby reducing the efficiency of their detection?

3: P19 and Fig. 4B: The authors could usefully describe data in panel C (RNH-/+) in this paragraph and not only in the Discussion. They might also add a small interpretation/conclusion, as to why over-expression of RNase H1 decreases the PLA signal in single thp2 delta and trf4 delta mutants.

4: It is unclear how the authors validated the efficiency of the S9.6 antibody. A control with/without in vitro treatment with recombinant RNase H might have been useful to assess the specificity of antibody S9.6.

5: As a comment, it is not clear that R-loops at the CAG repeats should be short (60 bp or less)? In reference 55 the authors suggested that R-loop accumulation at RNAPII promoter-proximal pause sites could be disrupted by the DRIP. This is understandable as nascent RNAs at promoter-proximal pausing sites are shorts, so R-loops should be inherently small. However, in the body of the gene, pausing of RNAP II might lead to formation of longer R-loops that would be detected by DRIP.

6: Line 161: Did the authors mean in absence of both RNase H1 and RNase H2?

7: P27: There is no mention of Trf4 in REF 72

Reviewer #3: In this article the authors study the impact of the THO complex and the TRAMP subunit Trf4 in maintaining CAG repeat stability and in the more general prevention of transcription replication conflicts. They show that deletion of the Thp2 and Mft1 members of the THO and of Trf4 and Rrp6, both involved in degradation of RNA in the nucleus, induces instability in a repeat-containing YAC. The mechanisms proposed for this instability are likely different, only the first seemingly involving increased formation of R-loops, at least as demonstrated by the suppression of the phenotype by RNaseH1 overexpression. Based on the increased RNAPII occupancy detected by ChIP in the region of the repeats the authors propose that excess transcription-replication conflicts might occur in mutants. Seemingly in agreement with this hypothesis, increased Proximity Ligation Assay signals generated by PCNA and RNAPII are detected in nuclei of mutant cells, which are suppressed by overexpression of RNaseH1 (but both in thp2∆ and trf4∆ cells, see below). The authors propose that instability in trf4∆ cells results from the titration of RPA by excess undigested RNA. Consistently, overexpression of RPA suppresses both the repeat-induced fragility and the increased PLA foci in trf4∆ cells.

This is a potentially interesting study, but not all of the conclusions are adequately supported and many important controls are missing. A few overstatements should be corrected or further supported. The possible titration of RPA by excess RNA is particularly interesting. A positive recommendation would certainly require providing the missing controls and addressing the concerns below.

Major points:

Fig. 1 and lines 203-207: The rate of appearance of 5-FOA resistant colonies in thp2∆ cells is 4 times that of wt cells in the absence of the CAG repeat and 4.7 in the presence of the repeat. According to this results the thp2∆ mutation affects equally the loss of the distal URA3 gene in the presence or the absence of the repeat. There is no significant exacerbation of the phenotype in the presence of the repeats as the authors say, which raises a serious concern about the interpretation of the results. 

Fig. 3B : in the DRIP experiment the RNaseH treatment is missing. This is absolutely required for demonstrating that the DRIP signal is generated by R-loops. Also it is not clear why the data are normalized to MMR1: is this a region deprived of R-Loops ? MMR1 forms R-loops according to the data from last Aguilera's paper (San Martin-Alonso et al.). The authors should justify the use of such a reference and repeat all DRIP experiments by including an RNaseH treated control which is a commonly used standard

Fig. 3C: Inferring from suppression by RNaseH overexpression that R-loops form but are not detected at the repeats is rather speculative. Can the authors include a positive control showing they can actually see DRIP signals in THO mutants ?

Fig. 3D: The RNAPII ChIP indicates increased polymerase occupancy but does not allow establishing if the signal represent stalled polymerases or increased transcription in the region (i.e. due to increased initiation). The authors should either prove there is no increased initiation or be more cautious in the interpretation of this result. Also, normalizing to ACT1 assumes there is no change for this gene. If this is the case, it should be shown. Also, the ChIP signals should be shown relative to a negative control (e.g. region with no polymerase).

Fig. 4A. PLA experiments are subject to artifacts. The authors should provide controls showing that no foci are seen in the absence of replication (e.g. in G1 arrested cells) or transcription (use of transcription inhibitors). Also, they should systematically count for each mutant the number of foci that are extranuclear, to evaluate the reliability of the technique. 

Fig 4BC. If trf4∆ induces genomic instability independently of R-loops, why overexpression of RNaseH1 suppresses formation of PLA foci in these cells ? This result does not appear to fit their model, according to what claimed in p.17 (lines 368-369). Also it appears incoherent with the fact that overexpression of RNase H1 does not suppress instability in trf4∆ cells (Fig 3D).

Fig 5A RPA ChIP should be controlled using a negative control (e.g. no antibody) and by showing (rather than normalizing to) the levels at ACT1. According to the model proposed, RPA should be sequestered by excess RNA in the trf4∆ context. If so, binding of RPA should be generally affected, and normalization to ACT1 would not be acceptable. If binding of RPA to ACT1 is not affected, the authors should explain why. 

Fig5B The authors should show that overexpression of RPA leads to increased levels of the protein and that its recruitment to CAG-70 is restored in these conditions.

---

## [Decision Letter · Decision Letter 2]

18 Nov 2022

Dear Dr Freudenreich,

Thank you for your patience while we considered your revised manuscript entitled "THO and TRAMP complexes prevent transcription-replication conflicts, DNA breaks, and CAG repeat contractions" for publication as a Research Article at PLOS Biology. This revised version of your manuscript has been evaluated by the PLOS Biology editors, the Academic Editor and the three original reviewers.

Based on the reviews (attached below), we are likely to accept this manuscript for publication, provided you address the data and other policy-related requests stated below.

In addition, we would like you to define the THO and TRAMP complexes in the abstract and to consider the following suggestion to improve the title:

"The RNA export and RNA decay complexes THO and TRAMP prevent transcription-replication conflicts, DNA breaks and CAG repeat contractions"

We expect to receive your revised manuscript within two weeks. 

*Published Peer Review History*

*Press*

Sincerely,

Ines

--

Ines Alvarez-Garcia, PhD

Senior Editor

PLOS Biology

on behalf of

Richard Hodge, PhD

Associate Editor

PLOS Biology

rhodge@plos.org

Fig. 1B-C; Fig. 2A-B; Fig. 3B-E; Fig. 4B-C; Fig. 5A-D; Fig. S1A-C; Fig. S2A-C; Fig. S3A-C and Fig. S4A-D

SPECIES INDICATED IN THE ABSTRACT? 

Please note that per journal policy, the model system/species studied should be clearly stated in the abstract of your manuscript.

We require the original, uncropped and minimally adjusted images supporting all blot and gel results reported in an article's figures or Supporting Information files. We will require these files before a manuscript can be accepted so please prepare and upload them now. Please carefully read our guidelines for how to prepare and upload this data: https://journals.plos.org/plosbiology/s/figures#loc-blot-and-gel-reporting-requirements

Reviewers' comments

Rev. 1: Frederic Chedin

The authors have addressed all my queries and I am satisfied at this point.

Rev. 2: David Tollervey

The authors have revised and improved the MS and I am happy to recommend publication.

Rev. 3:

The authors have satisfactorily addressed my concern, I can now recommend acceptance in PLoS Biology.

---

## [Editor Report · Decision Letter 3]

1 Dec 2022

Dear Catherine,

On behalf of my colleagues and the Academic Editor, Tanya Paull, I am pleased to say that we can accept your manuscript for publication, provided you address any remaining formatting and reporting issues. These will be detailed in an email you should receive within 2-3 business days from our colleagues in the journal operations team; no action is required from you until then. Please note that we will not be able to formally accept your manuscript and schedule it for publication until you have completed any requested changes.

PRESS

Kind regards, 

Richard

Richard Hodge, PhD

Associate Editor, PLOS Biology

rhodge@plos.org

PLOS
